# Distinct Cdc42 protein levels differentially regulate polarized growth and cell fusion in *Schizosaccharomyces pombe*

**Sajjita Saha[1], Aiswarya Sajeevan[1], Laura Merlini[1], Vincent Vincenzetti[2], Sophie G. Martin**[1,2]*

1 Department of Molecular and Cellular Biology, University of Geneva, Geneva, Switzerland,
2 Department of Fundamental Microbiology, University of Lausanne, Lausanne, Switzerland

* Sophie.Martin@unige.ch

## Abstract

The conserved Cdc42 GTPase is a key driver of symmetry breaking and polarized growth, forming zones of activity that locally recruit effectors to organize the cytoskeleton and polarize secretion. Here, we show that Cdc42 also functions in cell–cell fusion during *Schizosaccharomyces pombe* sexual reproduction, but concentrates at the fusion site through mechanisms distinct from those proposed in *Saccharomyces cerevisiae*. Notably, the *cdc42-mCherry*^SW allele, which is functional for cell polarization and has been used across organisms for dynamic studies, exhibits a strong fusion defect. These cells block fusion before cell wall digestion but after actin fusion focus formation, indicating that Cdc42 is required to translate the vesicle cluster into polarized cargo delivery. We trace the defect to instability of Cdc42-mCherry^SW and demonstrate that mating and cell fusion require higher Cdc42 protein levels than mitotic polarized growth. Remarkably, by constructing an allelic series driving Cdc42 expression over a 5-fold range, we discover that mitotic polarized growth responds linearly to Cdc42 protein levels, whereas mating exhibits a sharp switch-like response. We further trace this all-or-none response to pheromone-induced polarized growth. Thus, polarized growth in response to intrinsic or extrinsic cues exhibits distinct requirements to Cdc42 protein levels.

## Introduction

The small GTPase Cdc42 is a highly conserved Rho-family GTPase, which plays critical roles across eukaryotes. By coordinating multiple cellular processes and signaling pathways, Cdc42 is a key regulator of cell polarity [1,2]. It contributes to the organization of the actin cytoskeleton and vesicular trafficking [3], thus regulating cell migration, contractility, endocytosis, and secretion. It also regulates a host of signaling cascades, thus impacting the physiology of most organisms and cell types [4]. Even in simple single-cell yeast organisms, Cdc42 plays multiple essential roles,

**Data availability statement:** All relevant data are within the paper and its Supporting information files.

**Funding:** This work was supported by an iGE3 PhD salary award (https://www.ige3.unige.ch/) to SS, a Swiss Government Excellence Scholarship (https://www.sbfi.admin.ch/en/swiss-government-excellence-scholarships) to AS, and the University of Geneva (https://www.unige.ch/), the University of Lausanne (https://www.unil.ch), a Swiss National Science Foundation grant (https://www.snf.ch; 310030_207909) and ERC Advanced Grant (https://erc.europa.eu; SexYeast; 101019630) to SGM. The funders had no role in study design, data collection and analysis, decision to publish, or preparation of the manuscript.

**Competing interests:** I have read the journal's policy and the authors of this manuscript have the following competing interests: SM is a member of PLOS Biology's Editorial Board.

**Abbreviations:** DIC, differential interference contrast; FWHM, full width at half maximum; PAK, p21-Activated Kinases; sfGFP, superfolder GFP; SW, sandwich alleles.

in cell polarization [5,6], polarized secretion [7,8], cell division [9], cell fusion [10,11], and nuclear membrane repair [12].

Regulation and functional output of Cdc42 GTPase are best understood for its essential role in polarity establishment during mitotic growth in both fission and budding yeasts. Cdc42 is active when bound to GTP, which is promoted by guanine nucleotide exchange factors, and returns to the basal state upon hydrolysis to GDP, promoted by GTPase-activating proteins. For the emergence of cell polarity, a local patch of Cdc42-GTP can form spontaneously thanks to a conserved scaffold-mediated positive-feedback that promotes symmetry breaking [13–15]. A second, double-negative feedback promotes local Cdc42 activation, through membrane flow-induced removal of Cdc42 inhibitor from sites of polarity [16]. In wildtype cells, these positive and double-negative feedback loops are embedded in additional intrinsic cues that define where Cdc42 is activated. For instance, in fission yeast, microtubule-deposited factors help define the zones of Cdc42 activity at cell poles [17,18]. Upon GTP binding, the Cdc42 switch domains undergo conformational changes, recruiting effectors [19]. Some Cdc42 effectors are species-specific but many, including formins, the exocyst complex, and p21-Activated Kinases (PAK), are conserved across species [20,21]. In *S. pombe*, Cdc42 relieves autoinhibition of the formin For3 that assembles cables for directional transport by the myosin V Myo52 [22]; activates the exocyst complex for vesicle tethering at the plasma membrane [23,24]; and activates the essential PAK Pak1 [25], which also negatively feeds back to Cdc42 [26]. Collectively, these effectors promote the organization of a polarized actin cytoskeleton and vesicle secretion, converting the Cdc42-GTP patch into effective polarized cell growth [27].

Understanding of Cdc42 in vivo dynamics was allowed by the development of alleles tagged with either superfolder GFP (sfGFP) or mCherry fluorophore at the edge of the Rho-insert domain [28], a Rho GTPase-family specific alpha-helix [29]. These alleles, dubbed 'sandwich' alleles (SW), were extensively tested during mitotic polarized growth in fission yeast, with no reported phenotype [28], and the same tagging strategy was subsequently used in species across fungi, plants and animals [12,30,31]. Notably, these alleles allowed to understand that Cdc42-GTP exhibits slower mobility than Cdc42-GDP, leading to its accumulation at sites of activation [28]. They also showed that the overall fast mobility of both forms is required for the GTPase to repopulate polarity sites [32], where polarized secretion induces membrane flows that displace slow-mobile proteins such as Cdc42 GAPs [16].

Cdc42 GTPase also plays important roles at several stages of sexual reproduction. For sexual reproduction, cell polarization is re-wired to pheromone perception, allowing yeast cells to polarize growth in the direction of a mating partner that secretes peptide pheromones [33]. In fission yeast, P-cells release P-factor by canonical secretion, while M-cells secrete the lipid-modified M-factor through an ABC-transporter. Pheromone perception relies on cognate G-protein coupled receptors and downstream Ras-MAPK signaling cascade, which triggers a transcriptional program that promotes expression of all pheromone-signaling components and locks cells in sexual differentiation [34]. During early mating stages, Cdc42-GTP forms

 PLOS Biology

dynamic patches at the cell cortex, which recruit effectors, sample pheromones around the cell periphery and stabilize where optimal pheromone concentration is reached [35,36]. Cdc42 also promotes activation of the pheromone-MAPK signaling cascade, likely through the PAK Pak1 which promotes the activation of the MAPKKK [37–39].

A critical step during sexual reproduction is that of cell–cell fusion. In walled cells, such as yeasts, cell fusion requires two key steps: local digestion of the cell wall at the site of contact, followed by plasma membrane merging. Local cell wall digestion depends on the clustering of secretory vesicles containing cell wall hydrolytic enzymes at the site of cell–cell contact. This is mediated by the myosin V-based transport of the vesicles onto the actin fusion focus, a dedicated actin aster nucleated by formins [34]. In *S. pombe*, the mating-specific formin Fus1 assembles the actin fusion focus [40]. Consequently, deletion of *fus1* blocks cell fusion before cell wall digestion [41]. Cells are also blocked before cell wall digestion if condensation of the Fus1 focus fails [42,43]. Plasma membrane merging then relies on the 4-pass transmembrane domain protein Prm1. Its deletion blocks cell fusion after cell wall digestion, though active repair can rebuild the cell wall [44]. Interestingly, Cdc42 was shown to promote cell fusion in *S. cerevisiae*, functioning late in the fusion pathway after vesicle clustering but before cell wall digestion [10,11]. For this function, the GTPase forms a focus at the fusion site through direct interaction with a BAR domain complex Fus2-Rvs161 proposed to recognize the flattening of the plasma membrane upon partner cell contact [11,45–47]. In *S. pombe*, a possible role for Cdc42 GTPase in cell fusion has not been examined, although a function is likely given that deletion of the PAK Pak2 results in a partial fusion defect [48].

In this paper, we show that, in *S. pombe*, Cdc42 forms a focus at the fusion site like in *S. cerevisiae*, but through distinct mechanism. We report a specific cell fusion defect before cell wall digestion but after actin focus formation in cells carrying the *cdc42-mCherry*SW allele. We trace the origin of the defect to reduced levels of Cdc42-mCherrySW protein and show through several alleles that higher levels of Cdc42 protein are required for its roles in sexual reproduction than for vegetative polarized growth. By constructing a series of alleles probing a 5-fold range of protein expression, we further demonstrate that the requirements for Cdc42 are distinct for vegetative polarized growth, which responds linearly to Cdc42 levels, than mating, which exhibits an all-or-none response. We further show that this switch-like response also occurs for pheromone-induced polarized growth of single mating type cells.

## Results

### Cdc42 forms a focus and is active at the fusion focus

To examine Cdc42's role in cell fusion in *S. pombe*, we first looked at its localization. To detect total levels of Cdc42 by microscopy, we used Cdc42-sfGFPSW where the monomeric sfGFP is inserted at the edge of the Rho-insert domain after Q134. We previously reported that this sandwich fusion protein is fully functional during mitotic growth, in contrast to N- or C-terminally tagged alleles [28]. Cells carrying the *cdc42-sfGFPSW* allele are also largely mating and fusion proficient (see below and Fig 2A and 2B). We labeled the fusion site with Myo52-mScarlet, which accumulates at the actin fusion focus, and expressed the blue fluorophore mTagBFP2 under control of a P-cell promoter ($p^{map3}$), whose entry in the M-cell defines the time of fusion pore opening. Cdc42-sfGFPSW accumulated at the site of polarity during mating, and concentrated at the fusion site, reaching maximal levels at fusion time (Fig 1A). A significant GFP signal was also visible in the cytosol and vacuoles (see below). Using the CRIB-3GFP reporter to visualize active Cdc42-GTP [49], we similarly observed an increase in local Cdc42-GTP amounts at the fusion site, which disappeared post-fusion (Fig 1B). The slope of the increase in both total Cdc42 and Cdc42-GTP levels mirrored that of Myo52. Thus, Cdc42 local activity and amounts increase in a concordant manner at the fusion site until pore opening.

The similar profiles of Cdc42 and Myo52 fluorescence suggest that Cdc42 localization may depend on the actin fusion focus. To test this hypothesis, we probed whether Cdc42 recruitment relies on Fus1. For this and most subsequent experiments, we used homothallic (self-fertile) *h90* cells, in which both mating types are present in the population. In *fus1Δ* cells, Cdc42-sfGFPSW localized to the cell–cell contact site, but its distribution appeared wider than in WT, as shown by measurements of the full width at half maximum (FWHM) of its profile across the fusion axis (Fig 1C–1E). By contrast, Cdc42

PLOS Biology

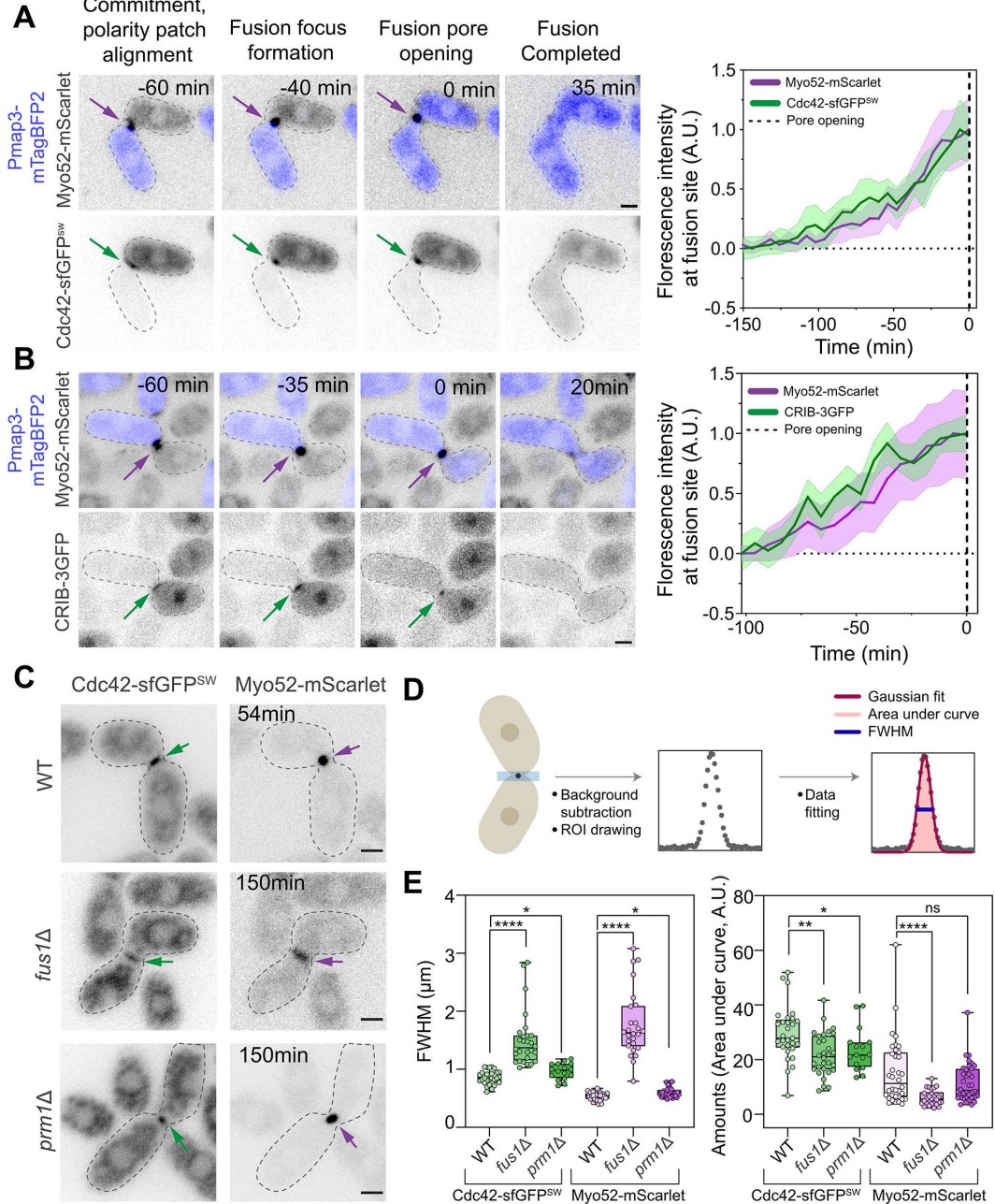

**Fig 1. Cdc42 concentrates and is active at the fusion focus. (A, B)** Time lapse of Cdc42-sfGFP^SW (A) and CRIB-3GFP (B) in *h−* cells crossed to *h+* cells expressing *p^map3^-mTagBFP2*. Both cells also express Myo52-mScarlet-I. Arrows point to the fusion site. Graphs on the right show average and SD of normalized intensities at fusion site. $n ≥ 15$. **(C)** Cdc42-sfGFP^SW and Myo52-mScarlet-I localization at cell–cell contact site (arrows) in *h90* (homothallic) WT, *fus1Δ* and *prm1Δ* strains. The indicated time is since cell pairing. The longer shapes of *fus1Δ* and *prm1Δ* mutants are due to extended growth due to fusion failure. **(D)** Scheme depicting the workflow of extracting FWHM and protein amounts (see methods). **(E)** FWHM and area under the curve values for WT ($n ≥ 29$), *fus1Δ* ($n ≥ 26$) and *prm1Δ* ($n ≥ 16$). In the boxplot, boxes comprise 25th to 75th percentiles with indicated median; whiskers extend to minimum and maximum value. Mann–Whitney test $p < 0.0001$: **** $p < 0.01$: ** $p < 0.05$: * $p > 0.05$: non-significant (ns). Scale bars = 2 μm. The data underlying this figure can be found in S1 Data.

distribution remained narrower in *prm1Δ*, which blocks cell fusion after fusion focus assembly. Thus, the change in Cdc42 distribution in *fus1Δ* is not an indirect consequence of the fusion failure but is due to absence of the actin fusion focus. As a control for presence or absence of the fusion focus, we measured the width of Myo52 distribution in the same cells, which confirmed that *fus1Δ* strongly perturbs Myo52 compaction, while *prm1Δ* has minor effects. Measurements of total Cdc42 levels showed a small reduction in both mutants (Fig 1E), perhaps because of the fusion block. We conclude that the focusing of Cdc42 at the fusion site requires the actin fusion focus, but that a broader localization of Cdc42 to the zone of cell–cell contact can occur independently.

## Cdc42-mCherry<sup>SW</sup> is impaired for cell fusion

The focal localization of Cdc42 at the fusion site is highly reminiscent of that described in *S. cerevisiae*, where Cdc42 forms a focus at the zone of cell contact before cell fusion [46]. In budding yeast, Cdc42 focus formation and function in cell fusion require its interaction with the amphiphysin-like protein Fus2 [11,46]. This interaction and Cdc42 function in cell fusion are specifically blocked in the *cdc42-137* and *cdc42-138* alleles, carrying D121A and D122A point mutations, respectively [11]. However, *S. cerevisiae* Fus2 does not exist in *S. pombe*. BLAST searches with *S. cerevisiae* Fus2 sequence only find homologs within the ascomycete *Saccharomycotina* and *Pezizomycotina* subphyla, but not the more basal *Taphrinomycotina* (the *S. pombe* subphylum) nor any non-ascomycete fungi. Furthermore, mutation of the Cdc42 interaction surface by introducing the homologous mutations *cdc42<sup>D121A</sup>* and *cdc42<sup>D122A</sup>* in *S. pombe* produced mutant cells that paired and fused like WT cells (S1A and S1B Fig). Thus, Cdc42, though forming a focus pre-fusion in both *S. cerevisiae* and *S. pombe*, does so through distinct mechanisms in the two species.

In parallel experiments, we observed that the *cdc42-mCherry<sup>SW</sup>* allele, while fully functional for cell polarization [28] and able to form cell pairs as efficiently as WT, exhibited a substantial defect in cell–cell fusion (Figs 2A, 2B, S1A, and S1B). The low fusion efficiency was also reflected in reduced iodine staining (which specifically stains spores) of *h90 cdc42-mCherry<sup>SW</sup>* cells (Fig 2C). Calcofluor staining revealed apparently intact cell wall in unfused *cdc42-mCherry<sup>SW</sup>* pairs (Fig 2D), indicating that fusion fails prior to cell wall digestion. Surprisingly, cells carrying *cdc42-sfGFP<sup>SW</sup>*, in which we examined Cdc42 localization above, were largely fusion-proficient, although we observed a small defect (Fig 2A and 2B). The distinct phenotypes of the two tagged strains are surprising given the similar size of sfGFP and mCherry and their identical site of insertion in Cdc42. To verify that the *cdc42-mCherry<sup>SW</sup>* phenotype was linked to the fluorophore, and not to a mutation in an unlinked genomic locus, we replaced mCherry by sfGFP in the *cdc42-mCherry<sup>SW</sup>* strain. The strains with swapped fluorophore showed efficient fusion, like the original *cdc42-sfGFP<sup>SW</sup>* strain (Fig 2E). We conclude that internal tagging of Cdc42, though functionally largely inconsequential for cell polarization [28], sexual differentiation, and cell pairing, specifically impairs cell–cell fusion before cell wall digestion. This phenotype is markedly enhanced with the mCherry fluorophore.

In the *cdc42<sup>SW</sup>* alleles, the fluorophores are inserted on the edge of the Rho-insert domain, distant from the effector-binding switch domains. In an attempt to identify Cdc42 surface residues involved in interactions potentially masked by the fluorophores, we introduced a series of point mutations on the Cdc42 surface near the mCherry insertion. However, none of these point mutations exhibited reduced iodine staining (which stains spores that form upon successful cell–cell fusion), nor did D121A and D122A (positioned on the other side of the Rho-insert domain) (S1C Fig). We only observed a strong mating defect upon removing the entire Rho-insert domain, which led to a distinct phenotype of complete sterility. These experiments, as well as the significantly lower functionality of Cdc42-mCherry<sup>SW</sup> than Cdc42-sfGFP<sup>SW</sup> suggested to us that their phenotype may not be fully explained by steric hindrance for effector binding.

Upon close examination of the *cdc42-mCherry<sup>SW</sup>* fusion phenotype, we observed that a few cell pairs exhibited spores in only one of the mating partners and apparent cell wall between the two partner cells (Fig 2A, arrowhead). This phenotype is reminiscent of cells lacking the Cdc42 effector Pak2, which show a partial fusion defect and transient fusion events, where a fusion pore opens but subsequently reseals [48]. Live cell imaging of *cdc42-mCherry<sup>SW</sup>* strains expressing

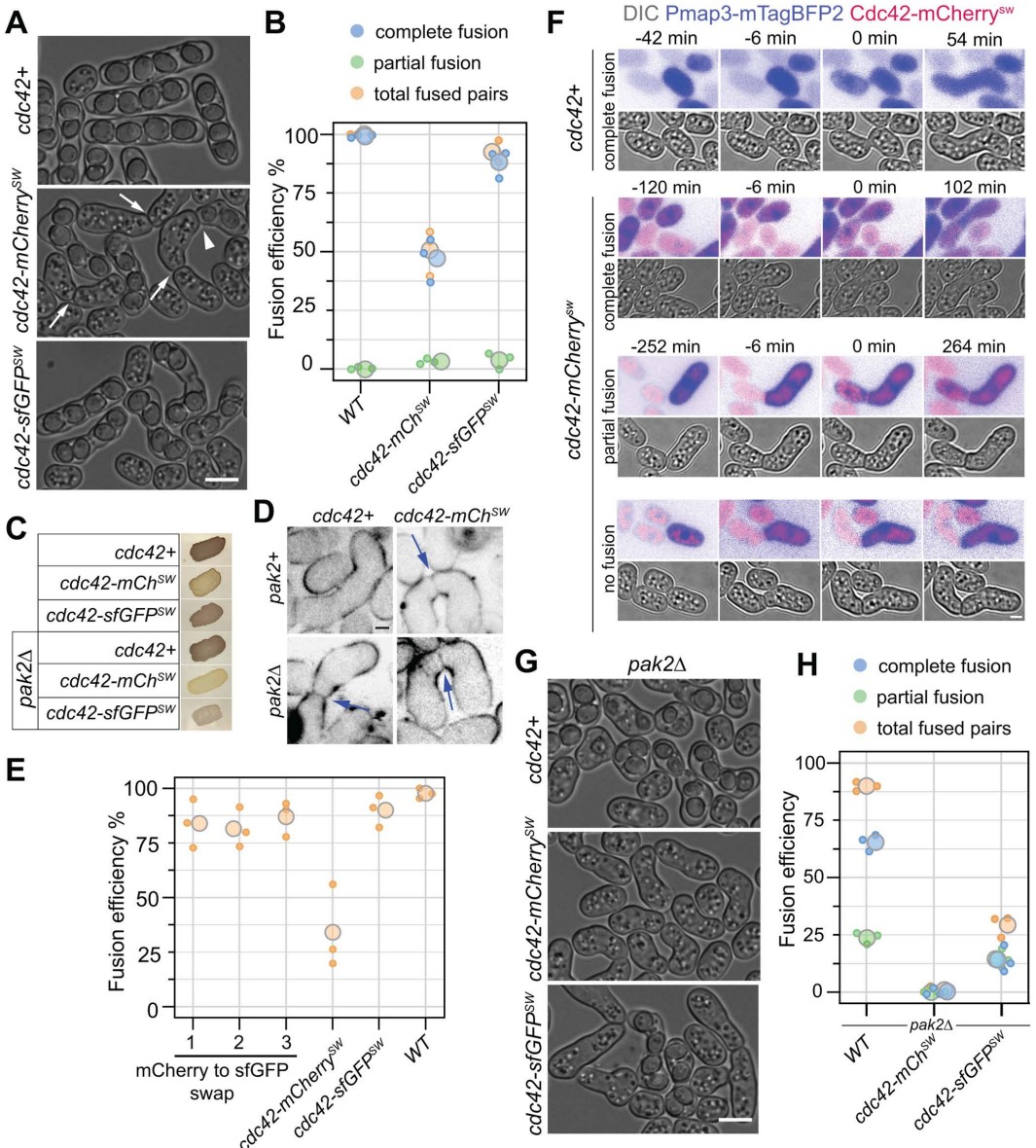

**Fig 2. Cdc42-mCherry^SW is impaired for cell fusion. (A)** Representative images showing fusion efficiency in *h90* WT, *cdc42-mCherry^SW*, and *cdc42-sfGFP^SW* strains. Arrows indicate failed fusion; arrowhead points to partial fusion with spores in only one partner. **(B)** Fusion efficiency plotted with values separated for complete and partial fusion for strains as in **(A)**, *N*=3, *n*≥139 cells per strain in each experiment. **(C)** Iodine staining of indicated *h90* strains. Dark color indicates presence of spores. **(D)** Calcofluor staining shows undigested cell wall (arrows) in WT, *cdc42-mCherry^SW*, *pak2Δ* and *cdc42-mCherry^SW pak2Δ* paired cells mated for ~24 h. **(E)** *cdc42-mCherry^SW* to *cdc42-sfGFP^SW* fluorophore swapped strains show WT like fusion efficiency. **(F)** Time lapse of *h90* WT and *cdc42-mCherry^SW* strains during cell fusion. The p^map3-mTagBFP2 expressed in P-cells indicates fusion time, as t0. In *cdc42-mCherry^SW* cells with partial fusion, mTagBFP2 redistributes to the M-cell but the cell wall remains undigested. In the bottom example, which does not fuse, time is indicated from start of imaging. **(G)** Representative images showing fusion efficiency in *pak2Δ* background, combined with *cdc42-mCherry^SW* and *cdc42-sfGFP^SW*. **(H)** Fusion efficiency plotted with values separated for complete and partial fusion for strains as in **(G)**, *N*=3, *n*≥102 cells per strain in each experiment. Scale bars=5 μm **(A, G)**; 2 μm **(D, F)**. In (B, E, **H**), small circles show individual experiments, large circles show averages. The data underlying this figure can be found in S1 Data.

mTagBFP2 in the P-cell under control of the $p^{map3}$ promoter indeed identified two types of fusion defects: a frequent instance of cells with complete fusion block, where mTagBFP2 remains in the P-cell, and rare cases of cells with partial fusion, where mTagBFP2 enters the M-cell, but the cell wall between the two partner cells appears intact in the DIC channel (Fig 2F). The similarity in *cdc42-mCherry*^SW and *pak2Δ* phenotypes led us to test whether the fusion defects observed in the *cdc42-mCherry*^SW mutant background arise from defective Pak2 activation. To test for this possibility, we conducted genetic epistasis experiments. If Cdc42-mCherry^SW failed to activate Pak2, double mutants should show no more severe phenotype than single *pak2Δ* mutants. By contrast, we found that *pak2Δ cdc42-mCherry*^SW double mutants were completely unable to fuse, with undigested cell wall at the cell–cell interface (Fig 2G–2H and 2C–2D). We also constructed *pak2Δ cdc42-sfGFP*^SW double mutants, which revealed that Cdc42-sfGFP^SW also has reduced functionality for fusion (Fig 2G–2H and 2C). Thus, *cdc42*^SW alleles exacerbate the fusion defect observed in *pak2Δ*. We conclude that the *cdc42*^SW fusion phenotype is not primarily due to impairment in Pak2 kinase activation.

## Cdc42-mCherry^SW compromises fusion but not actin fusion focus formation

Because Cdc42 accumulates on the fusion focus and is required for cell fusion, we examined its role at the fusion site. Given Cdc42's established roles in promoting actin cable assembly and exocyst activation during mitotic growth, we tested whether the Cdc42-mCherry^SW fusion protein causes any issue with vesicle accumulation at the fusion focus. We probed the localization of the formin Fus1 and type V myosin Myo52, major constituents of the fusion focus, the Rab GTPase Ypt3, which marks secretory vesicles, two exocyst subunits Exo70 and Sec6, and Prm1, a cargo of secretory vesicles delivered at the plasma membrane and necessary for membrane merging, each tagged with GFP. In wildtype cells, the exocyst decorates secretory vesicles in the focus, with Sec6 likely also decorating the plasma membrane [50]. We imaged each protein by time-lapse microscopy in *h90* strains. Since *pak2* deletion exacerbated the fusion defect, we also imaged *pak2Δ cdc42-mCherry*^SW double mutants. Each of these proteins localized correctly at the fusion site (Figs 3A and S2A), indicating that *cdc42-mCherry*^SW and *pak2Δ* do not cause gross abnormalities in organization of the actin cytoskeleton or polarized secretion.

Because changes in the compaction of the fusion focus and vesicle cluster can lead to fusion defects [42,43], we probed for more subtle changes by measuring the proteins' FWHM across the fusion axis, as in Fig 1D. To separate direct from indirect effects, we analyzed separately cell pairs that successfully fuse, which we measured 10 min before fusion, from those that failed. We note that those that managed to fuse took significantly longer time than WT to complete the fusion process, as indicated by a longer Myo52 persistence time (Fig 3B). With the exception of Prm1, we generally found little significant difference between their distribution in WT and *cdc42-mCherry*^SW mutants that successfully fused (Figs 3C and S2B; left graphs, first two lanes). Total protein amounts at the fusion site were also largely unchanged (Figs 3C and S2B; right graphs, first two lanes). Myo52 was also unchanged in mutant *cdc42-mCherry*^SW or *pak2Δ cdc42-mCherry*^SW cells mated with untagged WT ones, which successfully fuse (S2C Fig). By contrast, in cells that failed to fuse, whether single *cdc42-mCherry*^SW or double *pak2Δ cdc42-mCherry*^SW mutants, fusion focus and vesicle distribution was wider, with slightly reduced amounts of Fus1. We interpret this general wider distribution as a consequence of the fusion defect. The interesting exception to these general rules is the distribution of Prm1, which is significantly wider even in *cdc42-mCherry*^SW single mutants that managed to fuse (Fig 3A and 3C). Because Prm1 is delivered to the plasma membrane, its wider distribution may reflect a defect in translating the polarized organization of vesicles on the focus into a polarized cargo delivery at the plasma membrane. We note that Sec6 showed a similar wider distribution as Prm1 (S2B Fig), which may be similarly explained by its partial residence at the plasma membrane. Overall, these quantifications show that the *cdc42-mCherry*^SW mutant does not perturb the organization of the actin fusion focus but affects a downstream step in polarized secretion of the clustered vesicles.

## The fusion defect of *cdc42-mCherrySW* cells correlates with reduced Cdc42 protein levels

We were intrigued by the strong phenotypic difference between fusion-compromised *cdc42-mCherry*^SW and largely fusion-competent *cdc42-sfGFP*^SW cells. We made an initial observation that suggested that the two fusion proteins are

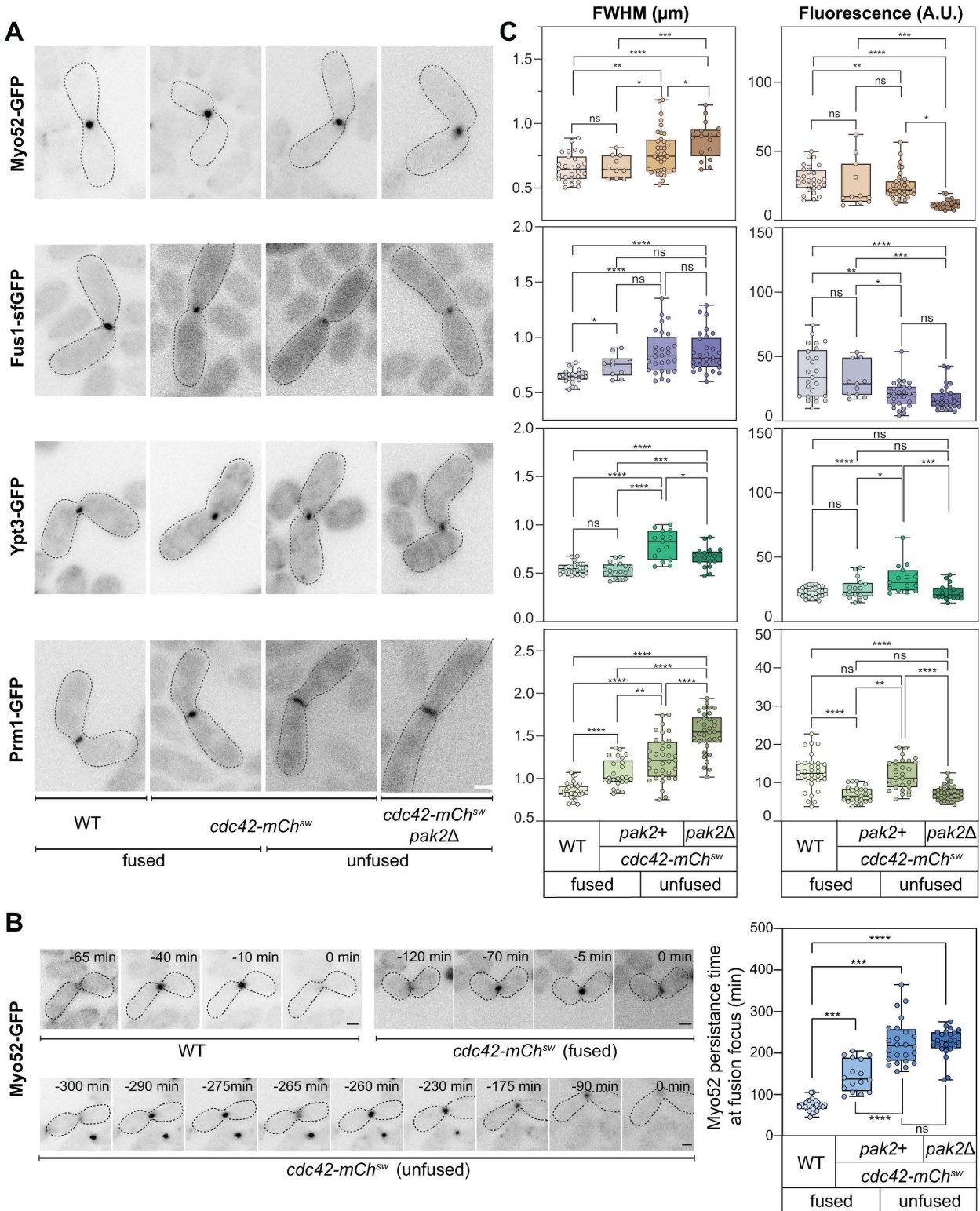

**Fig 3. Cdc42-mCherry^SW compromises fusion after actin fusion focus formation. (A)** Representative images of Myo52, Fus1, Ypt3 and Prm1 tagged with GFP in *h90* WT, *cdc42-mCherry^SW* and *cdc42-mCherry^SW pak2Δ* backgrounds, 10 min before fusion. Examples of fused and unfused pairs

are shown for *cdc42-mCherry$^{SW}$*. **(B)** Time lapse images showing persistence of Myo52 at the fusion focus in WT and *cdc42-mCherry$^{SW}$*, fused and unfused cases. Time 0 = disappearance of the Myo52 signal post-fusion or as cells give up (unfused case). The graph on the right shows the duration of Myo52 signal at the fusion site. WT n ≥ 33, *cdc42-mCherry$^{SW}$*(fused) n ≥ 16, *cdc42-mCherry$^{SW}$* (unfused) n ≥ 24, *cdc42-mCherry$^{SW}$* pak2Δ (unfused) n ≥ 24. **(C)** FWHM and area under the curve values plotted next to respective proteins from images as in **(A)**. Each dot corresponds to the measurement of a cell pair. In the boxplot, boxes comprise 25th to 75th percentiles with indicated median; whiskers extend to minimum and maximum value. For each protein: in WT background n ≥ 27, *cdc42-mCherry$^{SW}$*(fused) n ≥ 10, *cdc42-mCherry$^{SW}$* (unfused) n ≥ 15, *cdc42-mCherry$^{SW}$* pak2Δ (unfused) n ≥ 17. One way ANOVA $p < 0.0001$: **** $p < 0.01$: ** $p < 0.05$: * $p > 0.05$: non-significant (ns). Scale bars = 2 μm. The data underlying this figure can be found in S1 Data.

present at distinct levels in mating cells. While both Cdc42-mCherry$^{SW}$ and Cdc42-sfGFP$^{SW}$ decorate cellular membranes and accumulate at cell poles during mitotic growth, we observed additional substantial vacuolar fluorescence upon the nitrogen starvation that induces mating (Fig 4A). This vacuolar signal is likely the consequence of autophagy activation [51,52]. The vacuolar signal appeared stronger in the mCherry$^{SW}$ strain. Indeed, quantification of the vacuolar fluorescence intensity relative to whole cell intensity showed higher value for mCherry than sfGFP (Fig 4B). Conversely, we found that the fluorescence signal at cell poles (relative to whole cell intensity) was lower for mCherry than sfGFP (Fig 4B). While this suggests that the mCherry$^{SW}$ allele is more degraded, interpretation is difficult due to differences in stability and fluorescence of mCherry and sfGFP in the low pH of the vacuole. To probe Cdc42 levels more directly, we raised anti-Cdc42 antibodies and compared total Cdc42$^{WT}$, Cdc42-mCherry$^{SW}$, and Cdc42-sfGFP$^{SW}$ levels by western blotting in vegetative and mating conditions (Fig 4C). The up-shifted band in the tagged strains demonstrates specificity of the antibody. We first noted that Cdc42$^{WT}$ levels were unchanged in vegetative and mating samples (Fig 4C). Similarly, Cdc42-sfGFP$^{SW}$ levels were unchanged in the two conditions (despite the observed sfGFP vacuolar fluorescence during mating). By contrast, the full-length Cdc42-mCherry$^{SW}$ band was significantly reduced in mating extracts. Note that the Cdc42-mCherry$^{SW}$ strain also showed a strong degradation product in both growth conditions. This fragment, as well as a shorter one strongly present in mating extracts, is also detected by anti-RFP antibodies (see Fig 4E). Their sizes are consistent with a proteolytic cleavage in mCherry, about 30 aa from the Rho-insert domain. These observations suggest that the differential phenotypes of *cdc42-mCherry$^{SW}$* and *cdc42-sfGFP$^{SW}$* may stem from the lower levels of full-length Cdc42-mCherry$^{SW}$.

We next asked whether increasing the amounts of Cdc42-mCherry$^{SW}$ would rescue the fusion defective phenotype by over-expressing *cdc42-mCherry$^{SW}$* under the strong *p$^{act1}$* promoter, which led to about 3-fold increase in fluorescence (Fig 4D). We also under-expressed *cdc42-mCherry$^{SW}$* under the weaker *p$^{pom1}$* promoter. We verified by western blotting that the amounts of full-length Cdc42-mCherry$^{SW}$ were indeed increased or, respectively, decreased, during mating (Figs 4E and S3A). Over-expression of Cdc42-mCherry$^{SW}$ rescued the fusion defect, while under-expression made it worse compared to WT promoter *p$^{cdc42}$* (Fig 4F). We also quantified pairing efficiencies in these experiments. Because these were heterothallic cells confined in 2D on an agarose pad (instead of homothallic cells in 3D on a plate in most other experiments), the overall pairing efficiency is lower. In these conditions, the strain with Cdc42-mCherry$^{SW}$ expressed under endogenous promoter showed reduced pairing efficiency compared to WT. The pairing efficiency was rescued by overexpression with *p$^{act1}$* and decreased upon under-expression under *p$^{pom1}$* promoter. These results are consistent with the view that the low levels of Cdc42-mCherry$^{SW}$ during mating compromise cell fusion and also pairing.

### *cdc42* intron mutants with low protein levels and fusion defects

In a screen for *cdc42* viable alleles exhibiting mating defects, we identified three alleles which had no change in the *cdc42* coding sequence but point mutations in *cdc42* introns (Fig 5A). These mutants failed to produce spore progenies and were iodine-negative (Fig 5B). Insertion of a *cdc42* genomic rescue construct (which includes wildtype introns) rescued sexual reproduction, confirming that the defect is linked to the identified point mutations (Fig 5B). In each of these mutants, a point mutation maps to predicted intron acceptor site or branch point in either the first or second *cdc42* intron, suggesting they impair splicing, with likely consequence on protein levels (Fig 5A). Indeed, while we did not directly

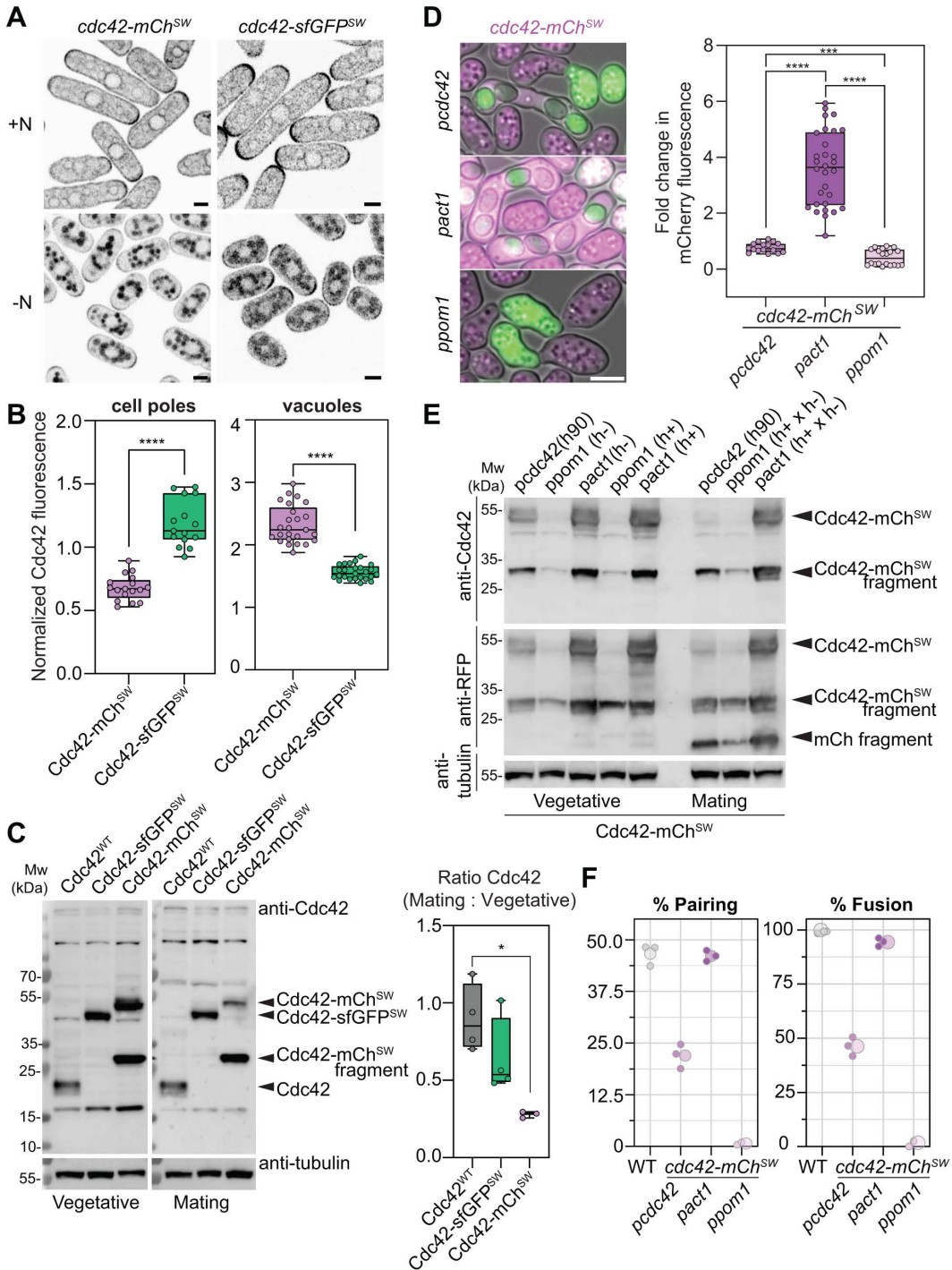

**Fig 4. The fusion defect of *cdc42-mCherry*<sup>SW</sup> cells correlates with reduced Cdc42 protein levels. (A)** Representative images of Cdc42-sfGFP$^{SW}$ and Cdc42-mCherry$^{SW}$ in cells during mitotic growth and nitrogen starvation (mating conditions). **(B)** Quantification of Cdc42$^{SW}$ abundance at cell poles ($n \geq 15$ cells) and vacuoles ($n \geq 30$ cells) during starvation, normalized to total fluorescence of the same cell. **(C)** Western blot of Cdc42 protein in indicated strains in mitotic and mating conditions. Quantifications show ratios of normalized Cdc42, Cdc42-sfGFP$^{SW}$ and Cdc42-mCherry$^{SW}$ amounts in mating over mitotic growth conditions, $n = 3$. Kruskal–Wallis test, $p < 0.05$: *. **(D)** Cdc42-mCherry$^{SW}$ over- and under-expression with $p^{act1}$ and $p^{pom1}$ promoters respectively. The $h+$ cells also express CRIB-3GFP, which serves as a cytosolic signal showing absence of fusion between partners. Graph shows intensity quantification of whole cells, $n \geq 16$. **(E)** Western blot of Cdc42-mCherry$^{SW}$ protein levels when expressed under

indicated promoters, probed with both anti-Cdc42 and anti-RFP antibodies. Individual strains during mitotic growth and mixed strains during mating are shown. **(F)** Pairing and fusion efficiency of *cdc42-mCherry$^{SW}$* heterothallic strains as in **(D)**, $N = 3$, $n \geq 90$ cells for each experiment. Mann–Whitney test $p < 0.0001$: **** $p < 0.001$: ***. Scale bars = 2 μm **(A)**; 5 μm **(D)**. The data underlying this figure can be found in S1 Data.

probe splicing, Cdc42 protein levels were vastly reduced on western blots (Fig 5C). During vegetative growth, each of these mutants was viable at 30 °C but showed temperature-sensitive growth defects at 36 °C (Fig 5D). At 25 °C, these mutants exhibited short cell length and increased width, but retained a rod shape, with a reduced aspect ratio (Fig 5F). Thus, very low levels of Cdc42 are sufficient to sustain viability, division and some level of polarized growth during mitotic proliferation.

During sexual reproduction, each of the three intron alleles showed reduced pairing efficiency, indicative of reduced pheromone signaling and/or polarized growth, and completely blocked cell–cell fusion (Fig 5E, 5G, and 5H). We attempted to overexpress the Cdc42$^{T195C}$ mutant protein under the strong *p$^{act1}$* promoter, which led to increased pairing and fusion efficiencies (Fig 5G and 5H), although the fused cell pairs showed an abnormal thin fusion neck (Fig 5E). This also restored thinner cell width (Fig 5F), but the strain remained temperature-sensitive for growth (Fig 5D), and we were unable to detect an increase in Cdc42 protein levels on western blots (Fig 5C). This suggests that Cdc42 levels were only mildly increased, perhaps due to very low levels of correctly spliced mRNA.

Together, these data further correlate low Cdc42 protein amounts with defects during mating and fusion, suggesting that these processes are more sensitive to Cdc42 levels.

### Mitotic cell polarization, mating, and cell fusion require distinct levels of Cdc42 protein

To directly test the importance of Cdc42 protein levels for its various cellular functions, we replaced the endogenous *cdc42* promoter with promoters of varied strengths to obtain different levels of protein expression. We initially used three weak promoters—*p$^{rga3}$*, *p$^{ypt1}$*, and *p$^{pom1}$*—and the strong promoter *p$^{act1}$*. We also constructed a control strain replacing the native promoter with *p$^{cdc42}$* (and selection marker). To obtain intermediate levels of Cdc42 expression, we subsequently inserted WT *cdc42* under control of the *p$^{pom1}$* promoter at a distinct genomic locus, yielding strains with two *cdc42* genes (*2x p$^{pom1}$* and *p$^{pom1}$ + p$^{rga3}$*). Western blots showed that Cdc42 levels increased in strains in the order *p$^{rga3}$ ≅ p$^{ypt1}$ < p$^{pom1}$ < p$^{pom1}$ + p$^{rga3}$ < 2X p$^{pom1}$ < p$^{cdc42}$ ≅ p$^{act1}$* (Fig 6A). Quantification of relative protein levels showed that the lowest expression was <20% that of endogenous promoter, with *p$^{pom1}$* and *2X p$^{pom1}$* at ~ 30% and ~ 60% of endogenous levels, respectively (Fig 6B). Note that loading of higher amounts of total proteins showed slightly higher levels of Cdc42 when expressed under *p$^{ypt1}$* than *p$^{rga3}$* promoter (S3B Fig).

All strains were viable and grew well at all temperatures, except *p$^{rga3}$-cdc42*, which was temperature sensitive at 36 °C (Fig 6C). To explore the consequence of changing Cdc42 protein levels, we measured the cellular dimensions. All strains formed rod-shaped cells with aspect ratio ranging from ~2.1 to ~3.6, but their length was progressively smaller and their width larger as Cdc42 protein levels were reduced (Fig 6D). Indeed, plotting cell length against Cdc42 levels showed a linear relationship, well approximated by a linear fit ($R^2 = 0.96$), indicating that the amount of Cdc42 determines cellular extension (Fig 6E). We observed a similar relationship between Cdc42 levels and cell width, but with a negative slope ($R^2 = 0.76$) (Fig 6F). We further used CRIB to estimate Cdc42-GTP levels at cell poles in a subset of strains (Fig 6G). Average measurement of CRIB levels at the two cell poles showed reduced amounts of Cdc42-GTP in the 2X *p$^{pom1}$-cdc42* strain relative to WT, and further reduction in the strains with lower Cdc42 levels (Fig 6H). We conclude that Cdc42 protein levels define the amounts of Cdc42-GTP and quantitatively define cellular dimensions. Notably, across 5-fold reduction in Cdc42 levels, the cells adopt a rod shape of decreasing aspect ratio. Thus, across this range, Cdc42 is limiting for polarized cell growth, but not for viability.

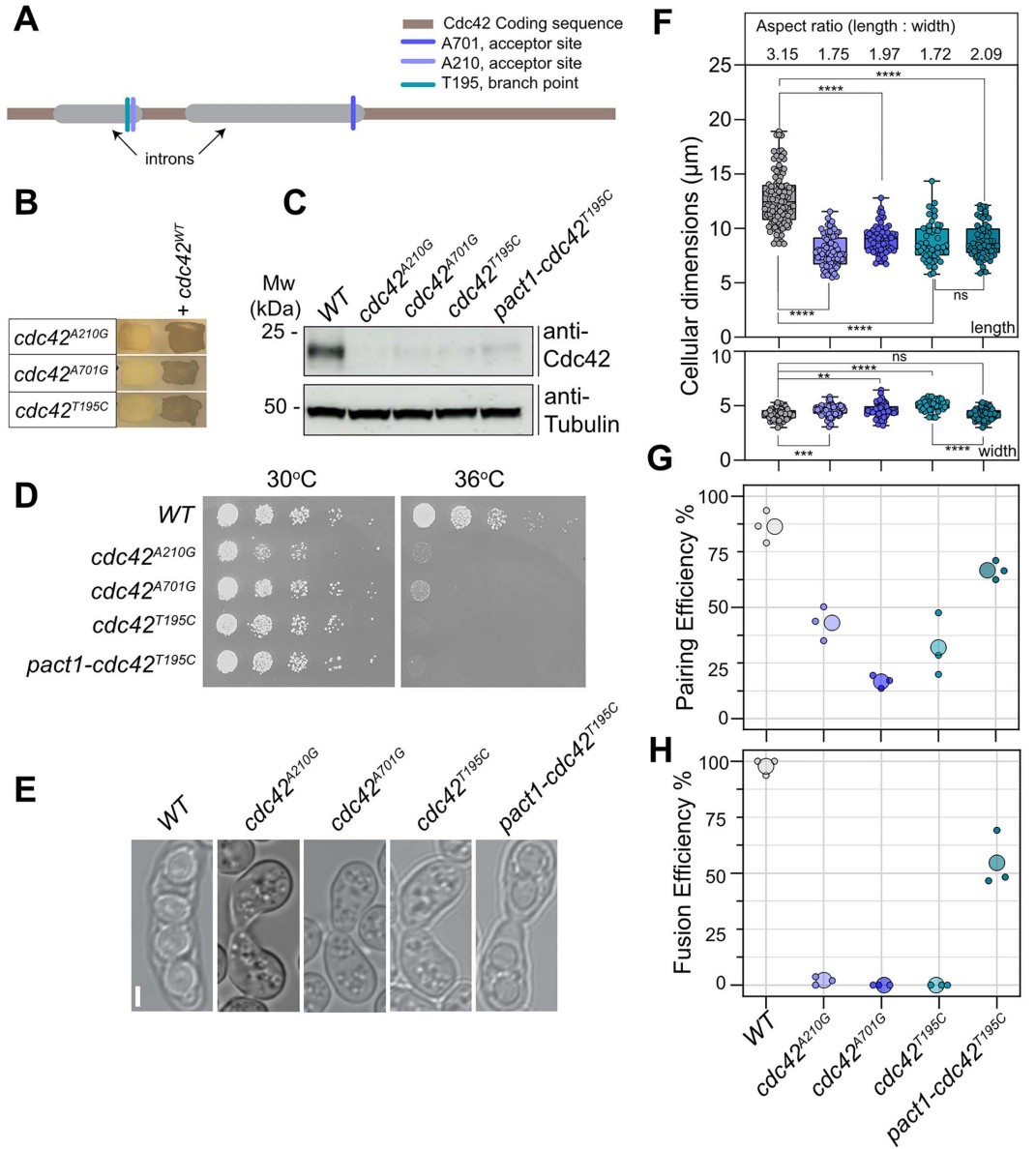

**Fig 5. *cdc42* intron mutants with low protein levels and fusion defects. (A)** Scheme of *cdc42* gene showing location of intron mutations. The *cdc42^A210G^* mutant also has a deletion of T503 in the second intron. **(B)** Iodine staining of indicated *h90* strains with or without *cdc42* genomic rescue construct. Dark color indicates presence of spores. **(C)** Western blot of Cdc42 protein levels during mating in intron mutants compared to WT. Similarly low levels are observed during mitotic growth (see S3 Fig). **(D)** 10-fold serial dilution of indicated strains on YE plates grown at 30 and 36 °C. **(E)** Representative images of fusion-defective paired cells of intron mutant strains; *p^act1^-cdc42^T195C^* shows fused pair with a narrow neck. **(F)** Cell length and width measurements, $n \geq 51$, each dot on boxplot corresponds to one cell. In the boxplot, boxes comprise 25th to 75th percentiles with indicated median; whiskers extend to minimum and maximum value. The average aspect ratio (length to width ratio) is shown on top. **(G)** Pairing and **(H)** Fusion efficiency of *cdc42* intron mutants. $N = 3$, $n \geq 96$ cells for each experiment. Mann–Whitney test $p < 0.0001$: **** $p < 0.001$: *** $p < 0.01$: ** $p > 0.05$: non-significant (ns). Scale bar = 2 μm. The data underlying this figure can be found in S1 Data.

During sexual reproduction, strains with Cdc42 expressed under $p^{cdc42}$ or $p^{act1}$ promoters at WT levels showed normal pairing and 100% fusion efficiency, whereas weak promoters ($p^{rga3}$, $p^{ypt1}$, $p^{pom1}$) prevented sexual reproduction (Fig 6I and 6J). In those strains, the very rare cell pairs that formed failed to fuse (Fig 6I and 6J). Interestingly, we observed a sharp

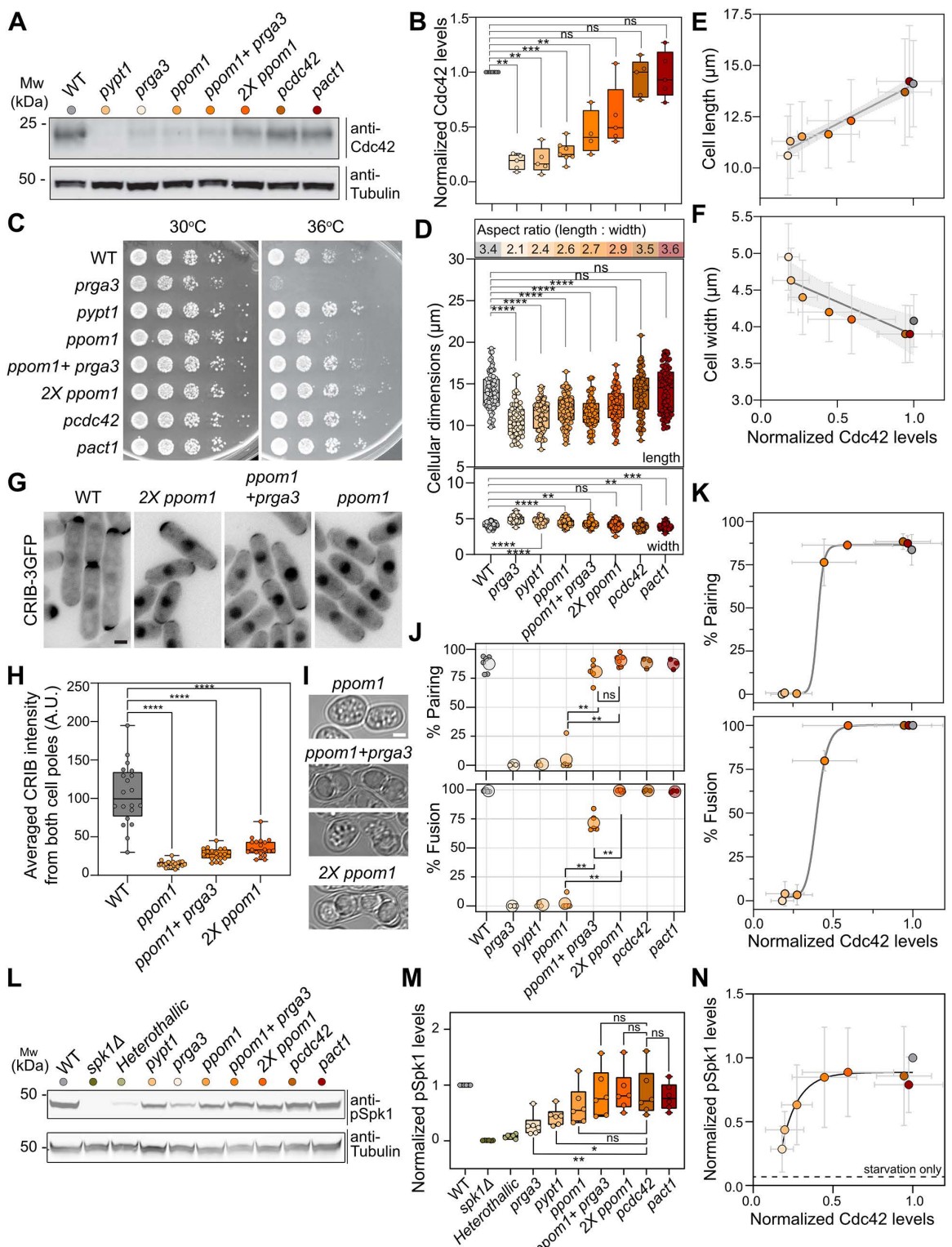

**Fig 6. Mitotic cell polarization, mating and cell fusion require distinct levels of Cdc42 protein. (A)** Western blot of Cdc42 protein levels when expressed under indicated promoters. Colored dots serve as legend to the graphs. **(B)** Quantification of Cdc42 levels from western blots as in **(A)**. **(C)** 10-fold serial dilution of strains expressing *cdc42* under indicated promoters on YE plates grown at 30 and 36 °C. Note temperature sensitivity of

$p^{rga3}$-cdc42. **(D)** Cell length and width of strains with varied Cdc42 expression levels, $n \geq 52$. The average aspect ratio (length to width ratio) is shown on top. **(E,F)** Linear relationship between cell length (E) or cell width (F) and Cdc42 protein levels, fitted with a linear regression model with 95% confidence interval. $R^2 = 0.95$ for cell length; 0.76 for cell width. Bars indicate standard deviations. **(G)** CRIB-3GFP in strains expressing cdc42 under indicated promoters. **(H)** Quantification of CRIB intensity at poles in strains as in **(G)**, $n = 20$, **(I)** Representative image of strains where cdc42 is expressed under control of $p^{pom1}$, $p^{pom1} + p^{rga3}$ or $2 \times p^{pom1}$ during mating. The arrowhead points to the narrow fusion neck. **(J)** Pairing and fusion efficiency of h90 strains with varied Cdc42 expression levels. $N = 3$, $n \geq 205$ cells for each experiment. Note that the fusion efficiency of alleles with low mating efficiency is based on observation of very few cell pairs. **(K)** Non-linear sigmoidal fit of pairing and fusion efficiency with Cdc42 levels (Hill coefficient $= 21.51 (R^2 = 0.99)$ for pairing efficiency and 12.06 $(R^2 = 0.99)$ for fusion efficiency). **(L)** Western blot of pSpk1 levels in indicated h90 strain backgrounds. Tubulin is used as loading control. **(M)** Quantification of pSpk1 levels as in **(L)**, Mann–Whitney test $p < 0.01$: ** $p < 0.05$: * $p > 0.05$: non-significant (ns). **(N)** pSpk1 levels as in (M) shown as function of Cdc42 levels, with exponential fit $(R^2 = 0.93)$. Mann-Whitney test $p < 0.0001$: **** $p < 0.001$: *** $p < 0.01$: ** $p < 0.05$: * $p > 0.05$: non-significant (ns). Scale bars $= 2$ μm. The data underlying this figure can be found in S1 Data.

step change between Cdc42 expression under $p^{pom1}$ and twice this level ($2X$ $p^{pom1}$), where the $2X$ $p^{pom1}$ strain was fully mating and fusion proficient. Intermediate Cdc42 levels, obtained by combining $p^{rga3}$ and $p^{pom1}$, yielded cells able to pair like WT but with a partial fusion defect (Fig 6J). In this strain, pairs that successfully fused exhibited an abnormally narrow fusion neck (Fig 6I). The relationship between Cdc42 protein levels and function during sexual reproduction was well modeled by a sigmoidal curve with high Hill coefficient and inflexion at ~40% Cdc42 levels (Fig 6K). Thus, the functions of Cdc42 in sexual reproduction are exquisitely sensitive to protein levels, with cell–cell fusion requiring slightly higher levels than sexual differentiation. Small variations in protein levels lead to an all-or-none type of response, very distinct from the linear response observed for mitotic polarized cell growth.

In summary, requirements for Cdc42 protein levels are distinct for its roles in viability, polarization, sexual differentiation and cell fusion. Notably, higher threshold levels are required for Cdc42 to promote cell–cell fusion.

## All-or-none mating decision shaped by pheromone communication

During mating, Cdc42 may contribute to activation of the pheromone-MAPK signaling cascade. To test whether changes in MAPK signaling explain the all-or-none cellular response to variations in Cdc42 protein levels, we tested the strains above for phospho-MAPK$^{Spk1}$, as a marker for activation of the pheromone-MAPK cascade [53]. Phospho-Spk1 was reduced in strains with lower Cdc42 levels (Fig 6L and 6M). However, sterile cells with $\leqq 30\%$ Cdc42 levels ($p^{pom1}$ and below) still activated the pheromone-MAPK cascade above the level observed upon only starvation in heterothallic cells (Fig 6N), suggesting sterility does not simply arises from complete signaling shut-off.

To test whether strains with low Cdc42 levels have the capacity to induce pheromone-MAPK signaling, we constructed the promoter series in heterothallic cells and tested them against WT partners. Except for the lowest-expressing $p^{rga3}$-cdc42 strain, heterothallic cells with reduced Cdc42 expression in either mating type mated and fused with WT partners (Fig 7A and 7B). Five-fold cdc42 under-expression under $p^{rga3}$ reduced mating and fusion efficiency in h+ cells and led to complete sterility in h− cells. The slightly higher levels of Cdc42 under $p^{ypt1}$ allowed mating but led to some fusion defects in h− cells. Thus, in crosses to WT partners, an all-or-none response is also observed, but at lower Cdc42 levels.

To simplify the system and test whether Cdc42 levels affect pheromone-dependent polarized growth, we studied the response of h+ cells with reduced Cdc42 levels to synthetic M-factor. Like the h90 strains, these h+ strains showed a linear relationship between Cdc42 levels and cell length during mitotic growth (S4A and S4B Fig). They all arrested with similar length upon nitrogen starvation (S4C Fig). In presence of 2,000 nM M-factor, all strains extended a single mating projection, except the $p^{rga3}$-cdc42 strain (Fig 7C). The relation of projection length to Cdc42 protein levels was similar to the response of these same h+ cells to WT h− partners (Fig 7D). 10-fold reduction of synthetic M-factor (200 nM) led to a shift in the response curve, where low-expressing cells did not grow, whereas higher-expressing ones did, with a sharp change between the $p^{pom1}$ and $2X$ $p^{pom1}$-driven levels (Fig 7C), similar to the relationship with the mating and fusion phenotypes in homothallic cells (Fig 7D). Thus, for both M-factor dosages, there is a sharp change in growth response across a small range of Cdc42 levels, with the position of the inflexion point dependent on pheromone dosage.

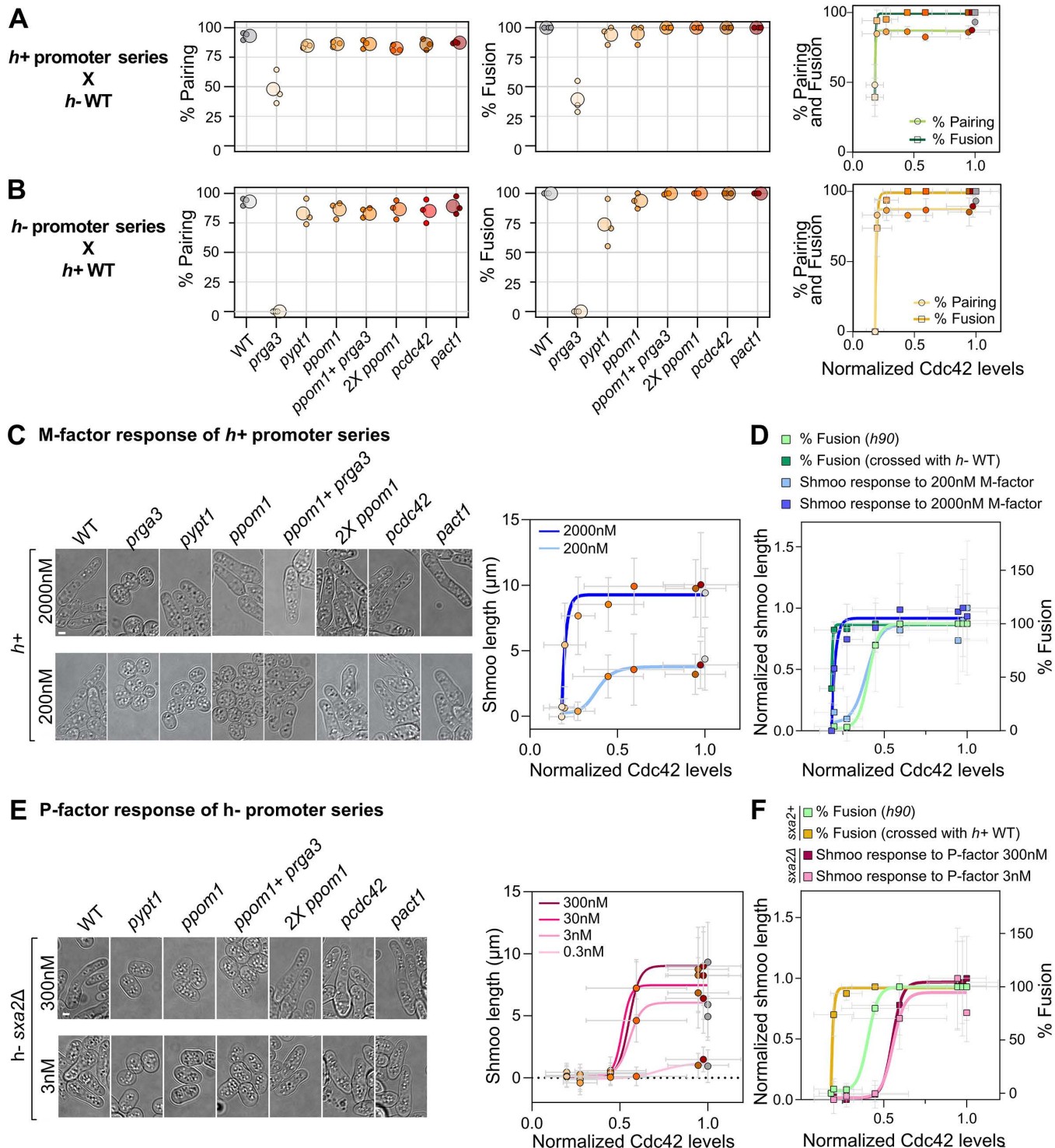

**Fig 7. Switch-like pheromone-induced polarized growth response to changes in Cdc42 levels. (A)** Pairing and fusion efficiencies of *h+* strains with varied Cdc42 expression levels, crossed with *h−* WT strains. *N* = 3, *n* ≥ 210 cells for each experiment. Non-linear sigmoidal fit of pairing and fusion efficiency with Cdc42 levels in shown on the right (Hill coefficient = 79.31($R^2$ = 0.95*)* for pairing efficiency and 67.32 ($R^2$ = 0.99*)* for fusion efficiency). **(B)** Pairing and fusion efficiency of *h−* strains with varied Cdc42 expression levels, crossed with *h+* WT strains. *N* = 3, *n* ≥ 223 cells for each experiment. Non-linear sigmoidal fit of pairing and fusion efficiency with Cdc42 levels in shown on the right (Hill coefficient = 81.93 (*$R^2$* = 0.99*)* for pairing efficiency

and 36.99 ($R^2 = 0.99$) for fusion efficiency). **(C)** Images of *h+* with varied Cdc42 expression levels exposed to 200 and 2,000 nM M-factor. Shmoo lengths as function of Cdc42 levels are shown on the right, with sigmoidal fit for 200 nM ($R^2 = 0.95$) and exponential fit for 2,000 nM ($R^2 = 0.87$) M-factor. **(D)** Overlay of the shmoo length of *h+* cells in response to 200 and 2,000 nM M-factor (from C) with the fusion efficiency of *h90* cells (from Fig 6K) and of *h+* cells crossed to WT (from A), as function of Cdc42 levels. **(E)** Images of *h− sxa2Δ* cells with varied Cdc42 expression levels exposed to 3 and 300 nM P-factor. Shmoo lengths as function of Cdc42 levels are shown on the right. Sigmoidal fit of response curves of P-factor concentrations 0.3 nM (Hill Coefficient = 5.32, $R^2 = 0.87$), 3 nM (Hill Coefficient = 12.55.32, $R^2 = 0.96$), 30 nM (Hill Coefficient = 18.04, $R^2 = 0.95$), and 300 nM (Hill Coefficient = 14.29, $R^2 = 0.99$). **(F)** Overlay of the shmoo length of *h− sxa2Δ* cells in response to 3 and 300 nM P-factor (from E) with the fusion efficiency of *h90* cells (from Fig 6K) and of *h−* cells crossed to WT (from B), as function of Cdc42 levels. Scale bars = 2 μm. The data underlying this figure can be found in S1 Data.

In parallel experiments, we tested the response of *h−* cells with reduced Cdc42 levels to synthetic P-factor. Because *h−* cells normally secrete the protease Sxa2 to degrade and shape P-factor gradients [54] and therefore do not respond to synthetic P-factor, we also deleted *sxa2* from these cells. We were, however, unable to obtain the *h− sxa2Δ p^{rga3}-cdc42* strain with lowest Cdc42 expression (see Materials and methods). We verified that, during mitotic growth, these cells exhibited a linear relationship between Cdc42 levels and cell length (S4D and S4E Fig). Upon nitrogen starvation, they arrested with similarly small cell length, though strains with lowest Cdc42 levels appeared slightly larger (S4F Fig). Interestingly, only cells with ≧60% of native Cdc42 levels (sharp change at 2X *p^{pom1}*) were able to extend shmoos in response to 3 nM P-factor, the minimal concentration to induce substantial outgrowth in *h− sxa2Δ cdc42+* cells (Fig 7E). Increasing P-factor concentration 10- or 100-fold also did not induce growth in cells with lower Cdc42 levels. Thus, the polarized growth response to homogeneous P-factor is also all-or-none, however, with an inflexion point at higher Cdc42 levels than for mating and fusion to either WT or self (Fig 7F).

In conclusion, our data show that polarized growth is differently sensitive to Cdc42 protein levels during the mating and mitotic cycles. While cells are relatively buffered to 5-fold changes in Cdc42 levels during mitotic growth, they are exquisitely sensitive to Cdc42 concentration over this range for pheromone-dependent growth and fusion responses, with pheromone dosage shifting the minimal required Cdc42 level.

## Discussion

In this study, we describe a critical role for Cdc42 GTPase in cell–cell fusion, demonstrating that this role is conserved across organisms, but its recruitment to the fusion site occurs through molecular connections that are evolutionarily divergent. In *Schizosaccharomyces pombe*, Cdc42 localizes in its active, GTP-loaded form at the fusion site, where it is concentrated by the actin fusion focus. In the *cdc42-mCherry^{SW}* mutant that we analyze in detail, we find no defect in the assembly of the actin fusion focus but show a late fusion defect, likely in the consequent local cell wall digestion. We further trace a main cause of the *cdc42-mCherry^{SW}* phenotype to a reduction in Cdc42 protein levels, providing three alternative genetic modifications of Cdc42 levels that all lead to cell fusion defects with minor effect on its roles during mitotic growth. Higher protein levels are required for the roles of Cdc42 in cell fusion and mating than in polarized cell growth and viability. Interestingly, Cdc42 protein levels produce a linear response for polarized mitotic cell growth but exhibit a step function for pheromone-dependent polarized growth during sexual reproduction.

### Role of Cdc42 GTPase for cell–cell fusion and specificity of *cdc42* mutant alleles

Cdc42 is a highly conserved small GTPase with a wide range of fundamental cellular functions. Previous work in *S. cerevisiae* had shown that functions of Cdc42 in cell–cell fusion are genetically separable from other functions [10,11]. We now reach similar conclusions in *S. pombe* using distinct mutant alleles that strongly perturb or completely block cell–cell fusion but have no or minor effects on the roles of Cdc42 for mitotic polarized growth.

The *cdc42* mutant alleles with fusion defects all have in common a reduction in protein levels, but vary in the reason behind this reduction. The *cdc42-mCherry^{SW}* allele provides the clearest separation of function, as these cells show no defect in cell polarization [28] but are strongly fusion defective. The main cause of the fusion defect is the reduction in

Cdc42-mCherry^SW protein levels during mating, which can be compensated for by over-expression. The *cdc42-sfGFP^SW* allele, tagged at the same location with a different fluorophore, leads to normal protein levels and is largely fusion-competent. However, its minor phenotype indicates that the presence of the tag also likely causes a slight hypomorphic protein function. We suspect that the difference in protein stability between the two alleles is linked to the difference in folding rates between the two fluorophores [55]. In the Cdc42-mCherry^SW-expressing strains, we observe a degradation product, with a size corresponding to a proteolytic cleavage in the Cdc42 Rho-insert domain just after the site of mCherry insertion. While this proteolytic cleavage is observed even during mitotic growth, the high rates of autophagy during sexual reproduction (triggered by nitrogen starvation [51,52]) may enhance it, leading to reduction in full-length protein levels. The high folding rate of sfGFP and ability to improve solubility of some fusion proteins [56] likely promotes Cdc42-sfGFP^SW folding, making it resistant to proteolytic cleavage.

The clearest demonstration that reduction of Cdc42 protein levels causes cell fusion defects is provided by the promoter exchange alleles. Notably, cells expressing Cdc42 at ~50% of native levels ($p^{pom1}$ + $p^{rga3}$) can grow as rods and efficiently form pairs during mating but show a partial fusion defect. A similar phenotype is seen in *h−* cells expressing Cdc42 at ~ 20% of native levels ($p^{ypt1}$) when crossed to WT *h+* cells. The phenotype of these alleles, which also exhibits short cell length during mitotic growth, is distinct from the normal cell dimensions of *cdc42-mCherry^SW* cells and is explained by the fact that Cdc42 levels are, in these cases, also reduced during mitotic growth. These alleles clearly show that higher protein levels are required for cells to fuse than to grow.

The intron alleles show a very severe fusion block but also exhibit more complex phenotypes. These cells have reduced mating efficiency and exhibit a temperature-sensitive growth defect. In this case, low protein levels likely result from post-transcriptional issues in splicing and possibly mRNA stability. The improvement in mating and fusion efficiencies upon expression under the stronger $p^{act1}$ promoter suggests that low levels indeed contribute to the phenotype. However, as we were unable to detect higher levels by western blot and the strain remained temperature-sensitive for growth, we suspect that there are additional causes to the observed phenotypes.

Together, these three types of *cdc42* alleles demonstrate that Cdc42 plays a key role in promoting cell fusion, which can be genetically separated from its function in promoting polarized cell growth, division, and viability.

## Evolutionary divergence in using Cdc42 GTPase for cell–cell fusion

Cdc42 GTPase is required for cell fusion in both *S. cerevisiae* and *S. pombe*. Together with the related GTPase Rac1 (absent in yeasts), it is also required for the fusion of myoblasts in mice and *Drosophila* [57–59]. Rac1 is further involved in the fusion of myeloid cells [60]. While these observations may at first glance suggest a common evolutionary origin, we find that the modes of Cdc42 GTPase localization/activation at the fusion site diverged between the two yeast species.

In *S. cerevisiae*, Cdc42's function in cell fusion requires its interaction with Fus2, a DH-BAR domain protein, which forms dimers with Rvs161 [11,45,47]. The model proposed is that Fus2-Rvs161 recognizes the flattening of the membrane upon contact between partner cells, leading to formation of a Cdc42 focus at this location [46]. In fission yeast, we show that Cdc42 does not localize through similar interaction, as there is no Fus2 homologue in this species, and the specific residues required for interaction in *S. cerevisiae* are dispensable for Cdc42 function in *S. pombe*. Instead, Cdc42 is concentrated at the fusion site by the actin fusion focus. Our previous work in mitotic cells showed that Cdc42 local accumulation results from its local activation, as the Cdc42-GTP form is less mobile than the inactive Cdc42-GDP form [28]. The coordinated increase of Cdc42 and CRIB signals at the fusion site suggests a similar mechanism may be at play during cell fusion. Future work should examine whether and how Cdc42 activators are concentrated by the actin fusion focus.

How does Cdc42 act on cell–cell fusion? By studying the *cdc42-mCherry^SW* phenotype, we show that cells fail to fuse before cell wall digestion but after formation of the actin fusion focus. Indeed, in *cdc42-mCherry^SW* cells that successfully fuse and can therefore be measured at comparable time as WT cells, Fus1, Myo52, and secretory vesicles marked by Ypt3 all form a focus of normal dimensions. If we consider *cdc42-mCherry^SW* mutants that fail to fuse, we measure a small

increase in the width of Fus1, Myo52, and secretory vesicle signals. However, this increase is unlike the widely spread signal observed in *fus1Δ* cells, in which the focus does not form [40], and more similar to that in *prm1Δ* cells, which block cell fusion after the function of the focus has promoted cell wall digestion [44]. Furthermore, we see no significant change in the distribution of Myo52 when *cdc42-mCherry^SW* or *cdc42-mCherry^SW pak2Δ* mutant cells are mated to WT cells. Thus, the minor changes in the width of the fusion focus in cells that fail to fuse are likely indirect consequences of the fusion failure, rather than a primary cell-intrinsic cause for failure. We conclude that, like in *S. cerevisiae* [11], Cdc42 acts a late stage after fusion focus formation.

The significantly wider distribution of Prm1 in *cdc42-mCherry^SW* cells (whether fusing or not) suggests that Cdc42 promotes the final steps of polarized secretion, after the cytoskeleton-dependent concentration of vesicles, for instance, promoting exocyst function, or perhaps enhances endocytic retrieval to achieve a highly polarized distribution of cargoes. At the molecular level, the observation that *pak2Δ cdc42-mCherry^SW* double mutants are fully fusion defective when single mutants only have partial defects demonstrates that Cdc42 does not solely function through the Pak2 kinase. Identifying the Cdc42 effectors for cell–cell fusion is an important target for the future.

In summary, the phenotypes of *cdc42* mutants suggest the GTPase exerts similar post-vesicle clustering functions to promote cell–cell fusion in budding and fission yeast. However, its accumulation at the fusion site occurs through distinct mechanisms.

## Distinct requirement of Cdc42 for mitotic growth and sexual reproduction

Interesting observations stem from our series of Cdc42 alleles expressed at levels ranging from 20% to 100% of endogenous levels. First, different Cdc42 amounts are required for different functions. Cells are viable with 20% of Cdc42, except at high temperatures; these cells are also clearly polarized, even if shorter than WT; however, they cannot mate. Cells can initiate sexual reproduction with 50%; they can only successfully complete fusion with 60%, although lower levels are permissible in one of the two mating partners. Whether regulatory mechanisms exist to ensure adequate amounts of Cdc42 protein is an open question. For instance, previous work showed that Cdc42 protein turnover is regulated by post-translational modifications during filamentous growth and in response to stress in *S. cerevisiae* [61,62]. While our data show that the steady-state levels of Cdc42 do not change during mitotic growth and mating, it is possible that turnover rates are different, especially considering the high levels of starvation-induced autophagy. In any case, our data show that a higher threshold of Cdc42 protein is required for sexual reproduction than for mitotic proliferation. This likely ensures that cell wall removal and fusion occur only when both partners are fully committed, thereby preserving the integrity of conjugating cells.

A second interesting finding is that the shape of the biological response to changes in Cdc42 levels is different for different Cdc42 functions. Notably, for sexual reproduction, we observe a sharp phenotypic change between cells expressing 30% and 60% of WT Cdc42 levels, with a slightly higher level required for cell fusion than cell pair formation. Thus, a~2-fold change in protein levels yields an all-or-none response in sexual reproduction. By contrast, over the same concentration range, the polarized growth output of Cdc42 is linearly determined by Cdc42 levels, which limit the pool of the active form, as observed by CRIB measurements. This linear response to Cdc42 levels suggests that, although likely critical for symmetry breaking, the scaffold-mediated positive feedback [13] and the membrane flow-dependent double-negative feedback [16] loops that enhance local Cdc42 activity do not dominate during mitotic growth. Perhaps linearity stems from control of Cdc42 activity by microtubule-delivered polarity factors [17,18].

By contrast, the sharpness of the none-to-all mating phenotype suggests an ultrasensitive response to Cdc42 activity during sexual reproduction. How does the strong Hill function come about during sexual reproduction? The first observation is that regulation of MAPK signaling does not appear to be the primary determinant of this response, as sterile cells show substantial MAPK phosphorylation. Ultra-sensitivity usually involves strong cooperativity or positive feedback. Given the two known feedback loops studied during mitotic growth [13,16], a simple hypothesis is that these underlie

symmetry-breaking for cell outgrowth in response to homogeneous pheromone, when polarized cell growth re-initiates after starvation and cells no longer rely on cell-intrinsic positional cues. Reduced exocytosis in cells with low Cdc42 levels may also lead to reduced levels of pheromone receptor at the cell surface. Thus, ultra-sensitivity is inherent to symmetry-breaking in response to pheromone signaling.

Interestingly, the position of the inflexion point marking the transition from no growth/sterility to growth/fertility varies between conditions and mating type. Analysis of the response of *h+* cells to M-factor is informative. Response to 200 nM of M-factor reproduces the sigmoidal curve of mutant cell mating, with an inflexion between cells expressing 30% and 60% of WT Cdc42 levels. Thus, low levels of Cdc42 compromise polarized growth in response to M-factor. Because these same cells efficiently mate with WT partners and extend shmoos to higher M-factor concentrations, this informs that WT cells likely produce >200 nM local concentrations of M-factor, whereas cells with <~50% Cdc42 levels release lower levels of M-factor. As hypothesized for pheromone receptors, the underlying reason may be a reduced delivery of M-factor transporter to the plasma membrane. Thus, low levels of Cdc42 also compromise M-factor release.

The analysis of the response of *h−* cells to P-factor is a bit more complicated. The observation that *h−* cells with low Cdc42 levels mate with WT, but not self indicates a role for Cdc42 in P-factor communication. We can also clearly conclude that low levels of Cdc42 compromise polarized growth in response to synthetic P-factor in *h− sxa2Δ* cells. However, a surprising observation is that, at all P-factor concentrations, *h− sxa2Δ* cells require higher levels of Cdc42 to grow than *h−* (*sxa2+*) cells to mate with either self or WT. The mating competence of the $p^{rga3}+p^{pom1}$ homothallic strain, but lack of shmoo growth in homogenous P-factor even at high concentration may be conferred by graded pheromone distribution, which reduces the need for symmetry breaking. Cdc42 may also affect the Sxa2 protease—perhaps modulating its secretion—to shape P-factor gradients and thus cellular response.

In conclusion, the inability of cells with low levels of Cdc42 to mate and fuse reflects a role of Cdc42 in the ultrasensitive growth response to pheromone, as well as a role in promoting pheromone release in both mating partners. Crosstalk between partner cells defines the specific Cdc42 levels at which cells transition from sterility to fertility.

## Materials and methods

### Strain construction

Standard genetic manipulation methods were used for *S. pombe* strain construction, by either transformation or tetrad dissection. All strains can be found in S1 Table, and plasmids in S2 Table. SW of *cdc42* are as described in [28]. Gene tagging (GFP tagging of *fus1*, *myo52*, *prm1*, *exo70*) was performed at endogenous genomic locus following PCR-based gene targeting at the 3′ end, yielding C-terminally tagged proteins, as described in [63]. GFP-Ypt3 protein was ectopically expressed in addition to the endogenous gene by transforming single integration vector pSM2366 (*3′UTRade6-pnmt41-GFP-Ypt3-hphMX-5′UTRade6*) targeting the *ade6* locus. Sec6-GFP tagged strains were produced by transforming linearized plasmid pSM3410 (pFA6a-Sec6-sfGFP-kanMX) which tags the endogenous locus of *sec6*.

Deletion of *fus1* was done by transformation of linearized plasmid pSM3134 (*pFA6a-5′UTRfus1-hphmX-3′UTRfus1*) which replaced *fus1* sequence with hphMX cassette. Deletion of *prm1* was obtained by PCR-based deletion by transforming PCR product amplified from pSM693 (pFA6a-hphMX6) with primers osm5913 (5′-GTTCTTTCGAAAATGTC TAAACAAAGAAATGCAAAA CCGTC TAACAC ATTTATTACAAGGCAGAGATCGAACAGCCTGTA CGGATCCCCGGGTTAATTAA-3′) and osm1816 (5′-GTGGAGGCGCTCCAACTCATTAGA TTTATTAAGCAGTTAAA TTATCC AAAATGAAAAATTTAGAATCT ATGTAGA CTTGAATTCGAGCTCGTTTAAAC-3′). pak2Δ was obtained from transformation of linearized plasmid pAV0220 (*pFA6a-5′UTRpak2-hphmX-3′UTRpak2*) which replaced *pak2* sequence with hphMX cassette. Fusion marker mTagBFP2 was introduced by transforming linearized plasmid pSM2955 (*5′UTRLys3-pmap3-mTagBFP2-bsdMX-3′UTRLys3*) into homothallic strains which integrates at *lys3* locus.

The screen yielding the *cdc42* intron mutants was based on the marker reconstitution strategy, essentially as described in [64]. Briefly, we transformed mutagenic PCR products amplified from pSM3024 (cdc42-his5c+) with primers osm1374

(GACGAAGCTCTTTCTAGAAGCGTAGT) and osm8302 (CGACTGCCTCCATAACTTCTGCATC) into a strain containing the other half of *his5* inserted after *cdc42*. Integrants were selected on EMM-ALU plates and screened by iodine staining for negative colonies. The *cdc42* gene was sequenced and the following point mutations identified: *cdc42^A210G^*: A210G in 1st intron and T503- in 2nd intron; *cdc42^A701G^*: A701G in 2nd intron and T1241C in 3'UTR; *cdc42^T195C^*: T195C in 1st intron (numbering from start of ORF). Mutants in *cdc42* coding sequence obtained in the screen will be described elsewhere. Rescue was performed by transformation of PmeI-linearized pSM3114, containing 2.7kb of *cdc42* genomic DNA starting 935bp upstream of START to 609bp downstream of STOP, leading to integration at *ade6* locus. Overexpression of intron mutant *cdc42^T195C^* was obtained by integration of plasmid *pFA6a-5'UTR^cdc42^-p^act1^-cdc42^T195C^-natMX-3'UTR^cdc42^* linearized by restriction enzyme MscI at the *cdc42* locus, replacing the native promoter.

The allelic series was obtained by first producing integrating plasmids containing cdc42 full gene expressed under promoters *p^ypt1^*, *p^rga3^*, *p^pom1^*, *p^cdc42^*, and *p^act1^*. Infusion cloning was used to combine promoter fragments with *cdc42* gene and the backbone which can be described as *5'UTR^cdc42^-p^promoter^-cdc42-natMX-3'UTR^cdc42^*. Plasmids were linearized with MscI and integrated at the endogenous *cdc42* locus, replacing the native promoter. 2X *p^pom1^* and *p^pom1^*+*p^rga3^* were obtained by integrating a second copy of *p^pom1^-cdc42* at the *ade6* locus using plasmid pSM3578 (*5'UTR^ade6^-p^pom1^-cdc42-hphMX-3'UTR^ade6^*). This was done in *h90*, *h+* and *h−* parental strains. The *h− sxa2Δ* strain series was obtained by crossing the *h−* series to *h+ sxa2Δ* cells. We could not obtain *h− sxa2Δ p^rga3^-cdc42* as *h− p^rga3^-cdc42* fails to mate with *h+* strains. We attempted to create *h− sxa2Δ p^rga3^-cdc42* by transformation, but observed diploidization of the transformants, hence we did not pursue this further.

## Cdc42 point mutations

To introduce the D121A and D122A mutations, as well as other point mutations in *cdc42*, we used the SpEDIT CRISPR protocol, as described [65]. Briefly, for each site, we constructed a pSLB-based plasmid for expression of a guide RNA, whose PAM sequence was chosen to be as close as possible to the targeted mutation and with good on- (>60) and off-target (>95) scores, for which we used Benchling tools (https://www.benchling.com/). This was co-transformed with a template for homologous repair obtained from annealing and polymerase filling-in of two overlapping 80-nucleotide oligonucleotides, carrying the desired mutation and mutagenizing the PAM site with a silent mutation. Transformants were sequenced to verify *cdc42* mutation.

## Growth conditions for imaging

For microscopy, fission yeast cells are grown in minimal sporulation medium with 15mM glutamate as main nitrogen source (MSL^Glutamate^), supplemented with appropriate amino acids. To evaluate cells during the mating process, either liquid or agar minimal sporulation medium lacking nitrogen (MSL-N) was utilized. Live cell imaging for mitotic phase was done on MSL^Glutamate^ agarose pads, and for mating phase, MSL-N agarose pads were used. All strains for imaging were grown at 30 °C unless differently indicated. Agarose pads were prepared following the protocol described in [66].

## Sample preparation for imaging

The precultures of *S. pombe* strains were grown in MSL^Glutamate^ at a temperature of 30°C with two overnight back dilutions ensuring they remain logarithmic. Subsequently, on the third day in morning, the cells (OD600 around 0.5–1) were transferred to MSL-N medium and subjected to three washing cycles with MSL-N to remove excess media. After washing, the cells were resuspended in 1ml of MSL-N to achieve a final OD600 of 1.5 and incubated at 30 °C for a duration of 3–4 hours, the exact length of which depended on the specific mating stage to be visualized. Following incubation, the cells were mounted onto MSL-N agarose pads containing 2% electrophoresis-grade agarose and covered with a coverslip sealed with VALAP (1:1:1 Vaseline/lanolin/paraffin). Samples were allowed to rest for about 30 min before imaging

for overnight movies on Nikon ECLIPSE Ti2 microscope at room temperature (controlled at ~23 °C). For imaging mitotic cells, growth medium is MSL$^{Glutamate}$ and cells are grown at 30 °C with two overnight back dilutions maintaining exponential growth. Sample preparation followed the above-mentioned steps were except the agarose pads were made with MSL$^{Glutamate}$ to sustain cell growth and division.

## Mating and fusion efficiency test

For assessing mating efficiency, strains were grown with two back dilutions in MSL$^{Glutamate}$ and washed with MSL-N three times on the morning of the third day. For each strain, the final OD600 value was adjusted to 1 in 1 ml of MSL-N medium. For heterothallic crosses (Fig 7), 0.5 ml from each mating type was then mixed. For all measures in Figs 5, 6, and 7, cells were starved at 30 °C in MSL-N for 2 hours and centrifuged at 3,000 rpm and the pellet was resuspended in 10μl MSL-N and spotted in MSL-N plates. Plates were incubated at 25 °C for 48 hours. For imaging the mating pairs, a tip was used to scrape off a small mass from the spot and mixed in 10 μl of MSL-N in an Eppendorf tube. Tubes were lightly vortexed to avoid cell clumps. 2 μl of that mixture was put on slide to capture differential interference contrast (DIC) images of mating pairs. For each mating efficiency test, a positive control of WT cells was done for comparison, and for each strain, mating efficiency was measured on three different days. For fusion efficiency calculations, both the total number of mating pairs and the number of fused mating pairs were quantified. Fused mating pairs were identified in DIC images as asci containing ascospores or as zygotes (mating pairs lacking a residual cell wall between them). The data collected were then used to calculate fusion efficiency as follows: fusion efficiency = (number of fused mating pairs/total number of mating pairs) × 100; pairing efficiency = (number of mating pairs × 2/total number of cells) × 100 for each cross. For measures in Figs 2, 4, and S1, cells were treated as above but mounted on MSL-N agarose pads as described for imaging, which leads to an overall lower pairing efficiency.

## Time-lapse imaging by epifluorescence microscopy

Live cell time-lapse imaging was performed using agarose pads sealed with VALAP on Nikon ECLIPSE Ti2 inverted microscope operated by the NIS Elements software (Nikon). Images were acquired using a Plan Apochromat Lambda 100 × 1.45 NA oil objective and recorded using Prime BSI express sCMOS camera from Teledyne Vision Solutions. Autofocusing was performed using the Nikon Perfect Focus System for long-term movies with multipoint acquisition on single z-plane (xy pixel size 0.65 × 0.65 μm). Exposure time and laser intensity were kept constant for every set of tagged protein imaging and each set subjected to 3-color imaging using fluorescence excitation at 405/488/561nm with the Lumencor SpectraX light engine (Chroma) in addition to DIC imaging with transmitted light.

## Image acquisition on confocal microscope

Cells were imaged on an airyscan-enabled Zeiss LSM980 confocal microscope fitted on an inverted Axio Observer 7 Microscope with an Airyscan2 detector optimized for a 63×/1.40 NA oil objective and laser lines 405 (Fig 2D) or 488 and 561nm (Fig 4A), with 1.5× digital zooming. Scanning was performed sequentially (x–y, 1,024 pixels × 1,024 pixels, where each pixel corresponds to approximately 0.048 μm), pinhole size was 1 AU, pixel depth was 16 bits, line averaging was set to 4, and scan speed of 8. ZEN Blue software was used for image acquisition and processing.

## Quantification of fluorescence intensity over time

All image visualization and processing were done with ImageJ/Fiji software (NIH, Bethesda, USA). Fluorescence intensity over time is measured for Fig 1A and 1B. Images were treated for background subtraction followed by alignment to correct for stage drift using an in-house ImageJ macro (https://github.com/Valentine-Thomas/FusionFocus-Mapping; Image_Processing/Centroid-detection_3channels [50]). CRIB-3GFP and Cdc42-sfGFP$^{SW}$ intensity over time were measured using

Myo52-mScarlet as a fiduciary marker in each cell pair where a 25 pixel × 25 pixel box was drawn on red channel displaying Myo52 signal and intensity values were recorded over time for both green and red channel until fusion pore opens. The time of fusion pore opening is identified by entry of the mTagBFP2 expressed in *h+* cell into the *h−* mating pair, and denoted as t0 in the image panel and the graph plot. Intensity values for each cell pair is normalized from 0 to 1, and mean values with SD are plotted over time on the x-axis.

## Measurement of FWHM and area under curve at fusion focus

Figs 1E, 3B, and S2 show FWHM values and corresponding area under curve values which we use to indicate size of protein distribution and total protein amounts at the fusion focus. Time-lapse imaging was done using epifluorescence microscope Nikon ECLIPSE Ti2 for each strain for 12 hours to capture the entire fusion process and images were treated for background subtraction followed by alignment to correct for stage drift, as above. For cells that fused, quantifications were done two frames (frame rate is 5 min) before t0 (i.e., 10 min before apparent fusion). For cells that did not fuse, we could not use fusion as reference time. As the protein signal at the fusion focus lasts for prolonged period and eventually disappears, we chose a frame when the signal is most prominent before its disappearance. The data fitting was done in semi-automated manner. After selection of the cell pairs, a 20-pixel-wide straight line was drawn across the fusion focus (depicted in 1D) and the fluorescence profile extracted (https://github.com/Valentine-Thomas/FusionFocus-Mapping; Spot_Width_Measure/line-profile.ijm [50]). A MATLAB program was used to fit the extracted profiles to a gaussian function: $f(x) = Ae^{\frac{-(x-\mu)^2}{2\sigma^2}}$, where, A is Amplitude, $\mu$ is mean and $\sigma$ is standard deviation. The area under the curve was calculated from: $\int_{-\infty}^{+\infty} Ae^{\frac{-(x-\mu)^2}{2\sigma^2}}$. FWHM was measured from $\sigma$ values of each fit with the formula: FWHM = $2\sigma\sqrt{2ln2}$.

## Cell dimension quantification

Cell dimensions are measured using automated segmentation pipeline that produced cell masks, in which we detected the major axis of the cell to determine cell length and minor axis to determine cell width [53]. DIC images were captured using the Nikon ECLIPSE Ti2 inverted microscope. For Figs 5F and 6D, we used asynchronous, exponentially growing cells grown in MSL$^{Glutamate}$. For Fig 7C–7F, we treated M and P-cells with P and M-factor respectively in MSL-N starved conditions. For Figs 5F and 6D, cell length and width are extracted from the masks and plotted directly. For Fig 7C–7F, the lengths of MSL-N starved cells (not incubated with pheromone) were also measured and an average value for each strain calculated. This value was subtracted from individual cell length values of pheromone-treated M- and P-cells to obtain the shmoo length. Sample sizes are indicated in the figure legend.

## CRIB-3GFP quantification at cell poles

Asynchronous, exponentially growing CRIB-GFP-expressing cells in MSL$^{Glutamate}$ were used for imaging by epifluorescence in Fig 6G and 6H. Images were processed for background noise as above. ROIs were drawn manually with the segmented line tool in ImageJ, where the lines were 10 pixels wide and covered the entire CRIB signal at the poles. Mean intensities were recorded from each cell pole and averaged for each cell. Sample size is indicated in the legend.

## Growth assay

Growth assay to test viability of *cdc42\** strains was done by growing them in YE (yeast extract) rich medium with two back dilutions and ensuring they are logarithmic (OD600 0.5–1.0). For each strain, OD was adjusted to 0.5 and 1:10 serially diluted in a 96-well plate. Strains were spotted on YE plates (5 μl per spot) with a 96-pin replica plater. Plates were incubated at 18, 25, 30, and 36 °C for 48 hours before plates were scanned. Mutant cells grew well at 18 and 25 °C, and only 30 and 36 °C are shown in Figs 5B and 6C.

## Myo52 persistence time

In Fig 3C, Myo52-GFP was imaged at 5-min interval by epifluorescence microscopy. The total time between first appearance of the Myo52 focus and its disappearance was recorded as persistence time for Myo52. For cell pairs that fused, this represents the length of the fusion process, from formation of the fusion focus until disappearance post-fusion. For cell pairs that did not fuse, this represents the lifetime of the focus before cells give up.

## Normalization of Cdc42-sfGFP<sup>SW</sup> and Cdc42-mCh<sup>SW</sup> levels at cell poles and vacuoles

Cells were grown as described above for imaging in mitotic (+N) and mating phases (−N) and imaged by Airyscan microscopy. For quantification, intensities at the cell poles were measured during starvation as described for CRIB-3GFP above. For vacuolar data, intensities from at least five vacuoles per cell were measured with a 5-pixel × 5-pixel box. For normalization (Fig 4B), we measured the total mean intensity for each cell in which pole and vacuolar signals were quantified and divide the cell pole and vacuole values by the mean cellular signal: Normalized Cdc42 fluorescence at cell pole = cell pole intensity value during mating/mean intensity value from same whole cell; Normalized Cdc42 fluorescence in vacuoles = mean vacuole intensity value during mating/mean intensity value from same whole cell. This provides a measure of the deviation from the mean at cell poles and vacuoles.

## P- and M-factor response assays

To observe M-cell shmooing in response to P-factor, we used the *h− sxa2Δ* promoter series. For M-factor response in P-cells, the *h+* promoter series was used. Strains were grown with two back dilutions in MSL<sup>Glutamate</sup> and washed with MSL-N three times on the morning of the third day. For each strain the final OD600 value was adjusted to 1 in 1 ml of MSL-N medium. Strains were washed three times with MSL-N medium and starved for 2 h. Agarose pads were created that contained P- or M-factor at indicated concentration (Fig 7C and 7E). Strains were incubated for 20 hours and DIC images were taken.

## Statistical analyses

Statistical analysis was conducted using GraphPad Prism version 10.0. The specific statistical tests applied for each analysis are detailed in the corresponding figure legends.

## Experimental conditions for cell culture and protein extraction for western blotting

Pre-cultures were incubated in 20 ml of MSL<sup>Glutamate</sup> or 20 ml of YE medium (for extracts from mating and vegetatively growing cells, respectively) at 25°C for 8 h with shaking at 180 rpm. In the evening, cultures were diluted to an [O.D.]600 nm of 0.02–0.04 in 80 ml of the respective media. By the next morning, vegetatively growing cells reaching an [O.D.]600 nm of 0.6–0.8 were used directly for extraction. The cells grown in MSL<sup>Glutamate</sup> upon reaching a similar [O.D.]600 nm were washed three times in 10 ml of MSL-N, resuspended in 30 ml of MSL-N at [O.D.]600 nm of 1, and incubated at 25 °C for 4–5 h (3–4 hours for the samples in Fig 7A) of mating before protein extraction.

For protein extraction, ice-cold 100% (w/v) trichloroacetic acid was added to cell cultures, reaching a final concentration of 10% (w/v), essentially as described [51]. Cells were pelleted by centrifugation at 1,000$g$ for 2 min at 4 °C, followed by a wash with 5 mL of pre-chilled (−20 °C) acetone. The pellet was then washed with 1 mL of lysis buffer (2% SDS, 10% glycerol, 50 mM Tris-HCl, 0.2 M EDTA, and complete protease inhibitor cocktail tablets (Roche, 11836145001)). For the samples in Fig 7A, the lysis buffer also contained PhosSTOP tablets (1 in 10 ml, Roche, 04 906 837 001) to inhibit phosphatase activity and preserve Spk1 phosphorylation. After washing, cells were resuspended in 400 µL lysis buffer. Cell lysis was performed by adding acid-washed glass beads (Sigma, G8772) and disrupting cells using a MagNA Lyser bead beater (Roche) for four cycles of 45 s at 6,500 rpm, with 30-s intervals between cycles. Lysates were centrifuged

at maximum speed for 5 min at 4 °C, and the supernatant was collected as the protein fraction. Protein concentration was quantified using the Bradford assay [67]. For further processing, samples were rapidly thawed and heated at 65 °C in 4× NuPAGE LDS sample buffer (Invitrogen, NP0007) for 15′. β-Mercaptoethanol (1 μL per 20 μL sample) was then added, and samples were incubated at room temperature for 10′ before either storage at −20 °C or immediate use for SDS-PAGE.

## Western blotting

Thirty μg of protein were loaded onto 4%–20% acrylamide gels (GenScript, M00656) and electrophoresed at 100 V for 2 hours using commercial Tris-MOPS-SDS running buffer (GenScript, M00138). Proteins were transferred onto a nitrocellulose membrane (Cytiva, 10600002) using a homemade transfer buffer (50 mM Tris base, 38 mM glycine, 1% SDS) supplemented with 20% ethanol at 100 V for 2 hours. Membranes were then blocked for 1 h in TBST containing 5% milk and incubated overnight in TBST with 5% milk and primary antibodies. Rabbit anti-Cdc42 antibody (raised against *S. pombe* Cdc42 peptides NH2-CARQHQHPLTHEQGER-CONH2 and NH2-CVAALDPPVPHKKKSK-COOH; Covalab) was used at a 1:2000 dilution. Mouse anti-RFP monoclonal antibody (Chromotek, mouse 6G6-100) and rabbit anti-pSpk1 antibody (Cell Signaling Technology, 4371), were used at a dilution of 1:2000. Tubulin was detected using HRP-conjugated mouse anti-alpha Tubulin antibody (Abcam, ab40742) at 1:2,000 dilution for 1h, except for samples in Fig 6L, where monoclonal mouse TAT-1 primary antibody (from Dr. Keith Gull, University of Oxford) was used at a 1:5,000 dilution overnight. Following primary antibody incubation, membranes were washed three times in TBST (15 min per wash) and incubated for 1 h with HRP-conjugated anti-rabbit (Promega, W401B) or anti-mouse (Promega, W402B) secondary antibodies, diluted 1:3,000 in TBST with 5% milk In Fig 7A, membranes were washed similarly after primary antibody incubation and were probed with IRDye 800CW anti-rabbit (LICORbio, 926-32213) or IRDye 800CW anti-mouse (LICORbio, 926-32212) secondary antibodies at 1:10,000 dilution in 5% milk for 1 h in dark. Membranes were then washed three times in TBST (15 min per wash). For chemiluminescence detection, membranes were washed with equal volumes of Peroxide Solution (Thermo Fisher, 1859701) and Luminol Enhancer Solution (Thermo Fisher, 1859698) and t signals were captured using the Amersham ImageQuant 800 imaging system (Cytiva). Fluorescent signals were captured using Sapphire FL biomolecular imager from Azure Biosystems with a 784/829 nm laser. All uncropped raw image files are available in S1 Raw Images.

## Image analysis and quantification for western blotting

Quantification of signal intensities was performed using ImageJ. Unprocessed TIFF files acquired from the imaging systems were opened in ImageJ, to measure the Mean Gray Value of protein bands. The region of interest was defined as the smallest area fitting the largest band in each row and was consistently applied to corresponding background measurements. To normalize intensity values, the recorded Mean Gray Values were inverted using the formula: $255 − X$, where $X$ represents the value obtained from ImageJ (with 255 being the maximum intensity in an 8-bit image). The net signal intensity for both protein bands and loading controls (Tubulin) was calculated by subtracting the inverted background intensity from the inverted band intensity. Normalized Cdc42 protein values were determined by first calculating the ratio of the net protein intensity to the net loading control intensity and then normalizing all values to that of the wild-type control on the same blot.

Numerical data for all graphs are provided in S1 Data.

## Supporting information

**S1 Fig. Cdc42 regulates cell fusion through distinct mechanisms in *S. pombe* and *S. cerevisiae*. (A)** Representative images of *h90 WT, cdc42^D121A, cdc42^D122A,* and *cdc42-mCherry^SW* during mating. The *cdc42^D121A* and *cdc42^D122A* alleles are equivalent to the fusion-defective *cdc42-137* and *cdc42-138* alleles in *S. cerevisiae* [11]. **(B)** Percentage of pairing and

mating efficiency in strains as in (A). **(C)** Alphafold predicted structure of Cdc42, with indicated switch domains binding effectors (yellow), location of mCherry$^{SW}$ and sfGFP$^{SW}$ insertion between Q134 and H135 (blue) and residues mutated to alanine by CRISPR, including D121 and D122 (green; others in cyan). The Rho-insert domain is the alpha-helix between D122 and Q134. **(D)** Iodine test for fusion and sporulation efficiency. Scale bar = 5 μm. Student $t$ test $p$-values; NS, non-significant ($p > 0.05$). The data underlying this figure can be found in S1 Data.
(PDF)

**S2 Fig. Exocyst component localization and bilateral phenotype of *cdc42-mCherry*SW. (A)** Representative images of Exo70 and Sec6 tagged with GFP in *h90* WT, *cdc42-mCherry*$^{SW}$, and *cdc42-mCherry*$^{SW}$ *pak2*Δ backgrounds, frames selected from time-lapses. Examples of fused and unfused pairs are shown for *cdc42-mCherry*$^{SW}$. **(B)** FWHM and area under the curve values plotted next to respective proteins. Each dot corresponds to measurement of a cell pair. In the boxplot, boxes comprise 25th to 75th percentiles with indicated median; whiskers extend to minimum and maximum value. For each protein: in WT background $n \geq 14$, *cdc42-mCherry*$^{SW}$(fused) $n \geq 7$, *cdc42-mCherry*$^{SW}$ (unfused) $n \geq 7$, *cdc42-mCherry*$^{SW}$ *pak2*Δ (unfused) $n \geq 17$. **(C)** Time lapse images of indicated *h90 myo52-GFP* (gray) strains crossed to untagged WT *h+* expressing mTagBFP2 (blue). FWHM values of Myo52-GFP in M-cells mating with untagged *h+* WT cells. Mann-Whitney test $p < 0.0001$: **** $p < 0.001$: *** $p < 0.01$: ** $p < 0.05$: * $p > 0.05$: non-significant (ns). Scale bar = 2 μm. The data underlying this figure can be found in S1 Data.
(PDF)

**S3 Fig. Alterations of Cdc42-mCherry$^{SW}$ levels and lower Cdc42 expression levels under *p*rga3 than *p*ypt1 promoter. (A)** Quantification of Cdc42-mCherry$^{SW}$ expressed under WT, $p^{pom1}$ and $p^{act1}$ promoters in vegetative and mating conditions from anti-Cdc42 western blots as in Fig 4E. **(B)** Western blots (from two independent experiments) with indicated amounts of total cell extracts for strains expressing Cdc42 under WT, $p^{rga,3}$ and $p^{ypt1}$ promoters. Note that the quantifications in Fig 6B were performed from signal from 30 μg total extracts. At 60 μg, a higher signal is observed for $p^{ypt1}$ than $p^{rga3}$ strains. The data underlying this figure can be found in S1 Data.
(PDF)

**S4 Fig. Cell length of heterothallic promoter series during mitotic growth and starvation. (A)** Cell length of *h+* strains with varied Cdc42 expression levels during mitotic growth (MSL + N), $n \geq 36$. **(B)** Linear relationship between cell length and Cdc42 protein levels in strains as in (A), fitted with a linear regression model with 95% confidence interval. $R^2 = 0.96$. Bars indicate standard deviations. **(C)** Cell length of *h+* strains with varied Cdc42 expression levels upon nitrogen starvation (MSL-N), $n \geq 35$. **(D)** Cell length of *h− sxa2*Δ strains with varied Cdc42 expression levels during mitotic growth (MSL + N), $n \geq 50$. **(E)** Linear relationship between cell length and Cdc42 protein levels in strains as in (D), fitted with a linear regression model with 95% confidence interval. $R^2 = 0.96$. Bars indicate standard deviations. **(F)** Cell length of *h− sxa2*Δ strains with varied Cdc42 expression levels upon nitrogen starvation (MSL-N), $n \geq 34$. The data underlying this figure can be found in S1 Data.
(PDF)

**S1 Table. Strains used in this study.**
(PDF)

**S2 Table. Plasmids used in this study.**
(PDF)

**S1 Data. Data set.**
(XLSX)

**S1 Raw Images. Uncropped western blots for** Figs 4, 5, 6, **and** S3. Reference to main figure panels is made in the figure. Please refer to the corresponding legend. Arrows indicate relevant bands.
(PDF)

## Acknowledgments

We thank Wanlan Li and Valentine Thomas (UNIGE) for help with data analysis code, and Thomas Kozusnik (UNIL) for generating the *cdc42^{D121A}* mutant.

## Author contributions

**Conceptualization:** Sophie G. Martin.

**Formal analysis:** Sajjita Saha, Aiswarya Sajeevan.

**Funding acquisition:** Sajjita Saha, Aiswarya Sajeevan, Sophie G. Martin.

**Investigation:** Sajjita Saha, Aiswarya Sajeevan, Laura Merlini, Vincent Vincenzetti.

**Methodology:** Sajjita Saha, Aiswarya Sajeevan, Laura Merlini.

**Project administration:** Sophie G. Martin.

**Supervision:** Sophie G. Martin.

**Validation:** Sajjita Saha.

**Visualization:** Sajjita Saha, Sophie G. Martin.

**Writing – original draft:** Sajjita Saha, Sophie G. Martin.

**Writing – review & editing:** Sajjita Saha, Aiswarya Sajeevan, Laura Merlini, Sophie G. Martin.

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
