## [Editor Report · Decision Letter 0]

15 Jan 2026

Dear Sophie,

Thank you for submitting your revised manuscript entitled "High levels of Cdc42 GTPase underlie an all-or-none decision to fuse" for consideration as a Research Article by PLOS Biology, following on from our initial decision in July.

Your revision has now been evaluated by the PLOS Biology editorial staff, as well as by an academic editor with relevant expertise, and I am writing to let you know that we would like to send your submission out for external peer review by the original reviewers at Review Commons.

Once your full submission is complete, your paper will undergo a series of checks in preparation for peer review. After your manuscript has passed the checks it will be sent out for review. To provide the metadata for your submission, please Login to Editorial Manager (https://www.editorialmanager.com/pbiology) within two working days, i.e. by Jan 17 2026 11:59PM.

Kind regards,

Richard

Richard Hodge, PhD

rhodge@plos.org

PLOS

---

## [Decision Letter · Decision Letter 1]

25 Feb 2026

Dear Sophie,

Thank you for your patience while we considered your revised manuscript from Review Commons entitled "High levels of Cdc42 GTPase underlie an all-or-none decision to fuse" for publication as a Research Article at PLOS Biology. Please accept my apologies for the delays that you have experienced during the peer review process. This revised version of your manuscript has been evaluated by the PLOS Biology editors, the Academic Editor and the original reviewers at Review Commons.

Please note that the initial Academic Editor has had to step down that this stage of the peer review process, so we have recruited an alternative academic editor to handle your manuscript at this point. Their comments are labelled as 'Comments from the Academic Editor' for full transparency. In addition, Reviewer #1 did not provide any specific comments, but recommended that we accept the manuscript.

Based on the reviews, I am pleased to say that we are likely to accept this manuscript for publication, provided you satisfactorily address the remaining points raised by the reviewer/academic editor. After discussions with the Academic Editor, we will not make the experimental request to delete the factors that assist tip-localization of Cdc42 in vegetative cells essential for the revision.

In addition, I would be grateful if you could please address the following editorial and data-related requests that I have provided below (A-D):

(A) We routinely suggest changes to titles to ensure maximum accessibility for a broad, non-specialist readership. In this case, we would suggest a minor edit to the title, as follows. Please ensure you change both the manuscript file and the online submission system, as they need to match for final acceptance:

“Distinct Cdc42 protein levels differentially regulate polarized growth and cell fusion in Schizosaccharomyces pombe”

(B) Thank you for already providing the source data for the figures in Table S3. According to journal convention, I would be grateful if this could be renamed to S1 Data.

(C) Please also ensure that each of the relevant figure legends in your manuscript include information on *where the underlying data can be found*, and ensure your supplemental data file/s has a legend (e.g. The underlying data can be found in S1 Data).

(D) Thank you for already providing the raw and uncropped gel images in Figure S5 and S6. However, I note that the blot for Figure S3B appears to be missing. I would also be grateful if you could please combine these images into a single supplementary file called S1 Raw Images.

We expect to receive your revised manuscript within two weeks.

*Published Peer Review History*

*Press*

Best wishes,

Richard

Richard Hodge, PhD

rhodge@plos.org

Reviewer remarks:

Reviewer #2: The authors have carefully and thoroughly responded to my comments and I am satisfied that this extremely interesting paper is ready for publication.

I have two very minor comments:

1. Figure 4 F - The authors have added revised test saying: "Pairing efficiencies in strains with Cdc42-mCherrySW expressed under endogenous or pact1 promoter were not significantly different from WT." This is surprising given that the figure shows a frequency of pairing for pcdc42 that is roughly 50% of wild type and pact1. Given the scatter it seems unlikely that this is just an issue of significance. Perhaps this is a typo or there is a more sophisticated explanation. In any event a careful reader would be mystified.

2. In the very interesting revised discussion about the ultrasensitive response to Cdc42 levels, I think the authors might remind the readers about why cells might have such an on-off switch for cell fusion. Specifically, cell wall removal would be lethal in the absence of a fully committed mating partner. Requiring high levels of cCdc42 function for shmoo formation, and even higher levels for fusion may help ensure the integrity of the conjugating cells.

COMMENTS FROM THE ACADEMIC EDITOR

This has evolved and improved as a result of the revisions. The authors have satisfactorily addressed my previous comments.

The new data in Fig. 7 are intriguing, and the speculations in the Discussion seem plausible. However, it remains the case that the basis for the different dose/response relation between vegetative and mating conditions has not be resolved. One potentially straightforward experiment would be to delete the factors that assist tip-localization of Cdc42 in vegetative cells, and see if that makes the dose-response more sigmoidal.

A minor point: how confident are the authors with regard to the concentration of M factor that the cells see? As I understand it, it is not at all trivial to know how much of a farnesylated peptide is actually available to the cells, so a nominal concentration of 200 nM may not be what the cells see.

---

## [Editor Report · Decision Letter 2]

2 Mar 2026

Dear Sophie,

On behalf of my colleagues and the Academic Editor, Daniel Lew, I am pleased to say that we can accept your manuscript for publication, provided you address any remaining formatting and reporting issues. These will be detailed in an email you should receive within 2-3 business days from our colleagues in the journal operations team; no action is required from you until then. Please note that we will not be able to formally accept your manuscript and schedule it for publication until you have completed any requested changes.

In addition, I realized that I forgot to ask you to update your competing interest statement during this round of minor revision, given that you are member of our editorial board. I took the liberty of adding the following sentence to the conflict of interest statement in the online submission form just in the interest of time, but please do feel free to revise the statement during the production process if you would like to edit it:

“I have read the journal’s policy and the authors of this manuscript have the following competing interests: SM is a member of PLOS Biology’s Editorial Board."

PRESS

Best wishes,

Richard

Richard Hodge, PhD

rhodge@plos.org

PLOS
